# Osteopontin binds ICOSL promoting tumor metastasis

Davide Raineri [1,2,9], Chiara Dianzani [3,9], Giuseppe Cappellano[1,2✉], Federica Maione [1,2], Gianluca Baldanzi [2,4], Ilaria Iacobucci [5,6], Nausicaa Clemente [1], Giulia Baldone[1,2], Elena Boggio [1], Casimiro L. Gigliotti [1], Renzo Boldorini[1,7], Josè M. Rojo [8], Maria Monti [5,6], Leila Birolo[6], Umberto Dianzani [1,7,10] & Annalisa Chiocchetti[1,2,10]

ICOSL/ICOS are costimulatory molecules pertaining to immune checkpoints; their binding transduces signals having anti-tumor activity. Osteopontin (OPN) is here identified as a ligand for ICOSL. OPN binds a different domain from that used by ICOS, and the binding induces a conformational change in OPN, exposing domains that are relevant for its functions. Here we show that in vitro, ICOSL triggering by OPN induces cell migration, while inhibiting anchorage-independent cell growth. The mouse 4T1 breast cancer model confirms these data. In vivo, OPN-triggering of ICOSL increases angiogenesis and tumor metastatization. The findings shed new light on ICOSL function and indicate that another partner beside ICOS may be involved; they also provide a rationale for developing alternative therapeutic approaches targeting this molecular trio.

[1] Dipartimento di Scienze della Salute, Interdisciplinary Research Center of Autoimmune Diseases—IRCAD, Università del Piemonte Orientale, 28100 Novara, Italy. [2] Center for Translational Research on Autoimmune and Allergic Disease—CAAD, Università del Piemonte Orientale, 28100 Novara, Italy. [3] Dipartimento di Scienza e Tecnologia del Farmaco, Università di Torino, 10125 Torino, Italy. [4] Department of Translational Medicine, University of Piemonte Orientale, 28100 Novara, Italy. [5] CEINGE Advanced Biotechnologies, 80145 Naples, Italy. [6] Dipartimento di Scienze Chimiche, Università di Napoli "Federico II", 80125 Napoli, Italy. [7] Laboratorio di Biochimica Clinica, Dipartimento di Scienze della Salute, AOU Maggiore della Carità, Università del Piemonte Orientale, Corso Mazzini 18, 28100 Novara, Italy. [8] Centro de Investigaciones Biológicas, Consejo Superior de Investigaciones Científicas (CSIC), 28040 Madrid, Spain. [9] These authors contributed equally: Davide Raineri, Chiara Dianzani. [10] These authors jointly supervised this work: Umberto Dianzani, Annalisa Chiocchetti. ✉email: giuseppe.cappellano@med.uniupo.it

  **1**

ICOSL (B7-H2, CD275) belongs to the B7 family and regulates the immune response by delivering costimulatory signals through ICOS, a surface receptor mainly expressed by activated T cells[1–4]. ICOSL is constitutively expressed by antigen-presenting cells (APCs) as well as some non-lymphoid cells, including several tumor-cell types[5,6]. The expression of ICOSL in non-lymphoid tissues, such as the brain, lungs, heart, kidney, liver, and gut, suggests that it regulates the activation of antigen-experienced effector/memory T cells. However, ICOSL is expressed at high levels in T helper follicular (TFH) cells, and ICOS deficiency has been associated with defective formation of lymphoid follicles in mice and in the development of common variable immunodeficiency in humans[7].

The main known function of ICOSL is triggering ICOS, which serves as a costimulatory molecule for T cells and, supports cytokine-driven polarization of T helper (Th) cells. Conversely, ICOS binding to ICOSL also triggers "reverse signaling" into the ICOSL expressing cell. In particular, triggering of ICOSL by a recombinant soluble form of ICOS (ICOS-Fc) inhibits adhesiveness and migration of human umbilical vein endothelial cells (HUVECs) and several tumor-cell lines, and treatment of mice with ICOS-Fc inhibits the development of experimental lung metastases in the B16 melanoma model[4,8]. Other effects of ICOSL triggering have been detected in dendritic cells (DC), whose treatment with ICOS-Fc not only inhibits cell migration and adhesion, but also modulates cytokine secretion and antigen cross-presentation in class I MHC molecules[9–11].

Osteopontin (OPN) is a phosphoprotein, secreted by several cell types, such as macrophages, DC and Th cells; it can function both as a matricellular protein and a cytokine mediating several biological functions. These include migration, adhesion, activation of inflammatory cells, and modulation of T cell activation supporting differentiation of proinflammatory type 1 (Th1) and type 17 (Th17) Th cells[12]. It is highly expressed in several types of tumors, being secreted by both tumor and microenvironment cells, and is implicated in promoting tumor invasion and metastatic dissemination[13]. Importantly, current research, indicates that OPN inhibition would be a good therapeutic approach to metastatic disease. In preclinical models, OPN knockdown using RNAi, aptamers, or antibodies, have shown to have active roles in cancer treatment. Nevertheless, only a small number of these findings translate into clinical practice[14].

OPN's pleiotropic activities are partly due to its capacity to interact with multiple ligands, including several cell-surface receptors, namely several integrins and CD44, calcium, and heparin. OPN's biological functions are also influenced by post-translational modifications, such as phosphorylation, glycosylation, and protein cleavage mediated by thrombin and metalloproteinases[15].

OPN is an intrinsically disordered protein (IDP), which challenges the traditional structure-function paradigm of biologically relevant proteins[16]. Indeed, OPN lacks a well-folded crystallizable structure, but behaves either as an extended and flexible polypeptide or as a globular protein. This confers to OPN the ability to adopt different functional structures by folding upon binding and to interact with multiple binding partners[17]. Thrombin cleaves OPN in the middle of the molecule, near to an RGD motif, and generates two fragments with slightly different functional activities[12].

Recent findings depict a functional network between ICOS, ICOSL, and OPN. Firstly, all these molecules support Th17 cell responses. Secondly, they are involved in cell migration, which is induced by OPN and inhibited by ICOS-mediated ICOSL triggering[10]. Thirdly, they are involved in the bone metabolism, since OPN is a key bone component produced by osteoblasts, whereas ICOS-mediated triggering of ICOSL expressed by osteoclasts

(OC) inhibits OC differentiation from monocytes and bone resorption activity of mature OC in vitro, and the development of osteoporosis in vivo[18].

A prior study showed that HUVECs treatment with ICOS-Fc inhibits ERK phosphorylation induced by OPN, but not that induced by ATP[8]. More recent work found that ICOS-Fc inhibits HUVEC tubulogenesis induced by OPN, but not that induced by VEGFα[19]. Collectively, these data suggest an intersection between ICOS/ICOSL and OPN functions, and lead to this study aimed at investigating the crosstalk among this molecular trio.

Here, we show that OPN binds ICOSL at a different domain than the one used by ICOS. Activation of ICOSL by OPN induces cell migration in vitro, angiogenesis, and tumor metastatization in vivo.

## Results

**Expression of ICOSL is required for OPN-mediated cell migration and is blocked by ICOS-Fc.** The effect of ICOSL triggering by ICOS-Fc was evaluated in OPN-induced migration of several human tumor-cell lines expressing high (ICOSL$^{high}$) or low (ICOSL$^{low}$) levels of ICOSL (Supplementary Figs. 1a, b, respectively). In parallel, the effect on migration induced by 20% FBS was assessed. Results showed that both OPN and FBS induced substantial migration of the ICOSL$^{high}$ cell lines (M14 and JR8), which was inhibited by ICOS-Fc (Fig. 1a). By contrast, ICOSL$^{low}$ (PCF2 and A2058) cell migration was induced by FBS but not by OPN (Fig. 1b). Treatment with ICOS-Fc did not affect the migration of the ICOSL$^{low}$ cell lines in any setting.

Since these data suggest that ICOSL expression is required for OPN-induced migration, the study assessed whether overexpression of ICOSL in the A2058 cells conferred migration capability in response to OPN. A2058 cells transfected with a plasmid encoding for ICOSL (Supplementary Fig. 1c) acquired migratory capability in response to OPN, and this migration was inhibited by ICOS-Fc (Fig. 1c).

To assess the role of ICOSL in OPN-induced migration in a different and non-tumor-derived cell model, HUVECs, which are known to express ICOSL, were tested. To silence ICOSL, HUVECs were transfected with two different siRNA targeting ICOSL (siRNA1 and siRNA2) or scrambled siRNA (siRNAscr) as control. Real-time PCR confirmed statistically significant silencing of ICOSL in cells transfected with siRNA1 and siRNA2, compared with siRNAscr (Fig. 1d). Differentially from siRNAscr, HUVECs silenced for ICOSL did not respond to OPN while maintain their response to VEGF (Fig. 1e).

Taken together, these findings show that OPN-induced migration requires the presence of ICOSL.

To investigate which portion of ICOSL is required for responsiveness to OPN-induced migration, OPN pro-migratory activity was assessed in not expressing ICOSL-HeLa cells (cervical cancer cell line), in comparison with HeLa transfected for full-length ICOSL (ICOSL$_{WT}$), or an intracellular tailless mutant (ICOSL$_{TL}$), or a mutant composed of the transmembrane and intracellular portions plus a poly-histidine sequence substituting the extracellular portion (ICOSL$_{IC}$) (Fig. 1f). Expression of ICOSL was assessed by flow cytometry using an anti-ICOSL mAb for ICOSL and ICOSL$_{TL,}$ and an anti-poly-His PE-conjugated mAb for ICOSL$_{IC}$ (Fig. 1g). We found that only HeLa transfected for full-length ICOSL (ICOSL$_{WT}$) molecule gained the ability to migrate toward OPN, whereas those transfected with either ICOSL$_{IC}$ or ICOSL$_{TL}$ did not. By contrast, all these cell lines displayed a similar migration in response to FBS (Fig. 1h).

**OPN binds the extracellular portion of ICOSL.** These data suggest that OPN triggers pro-migratory signals through ICOSL

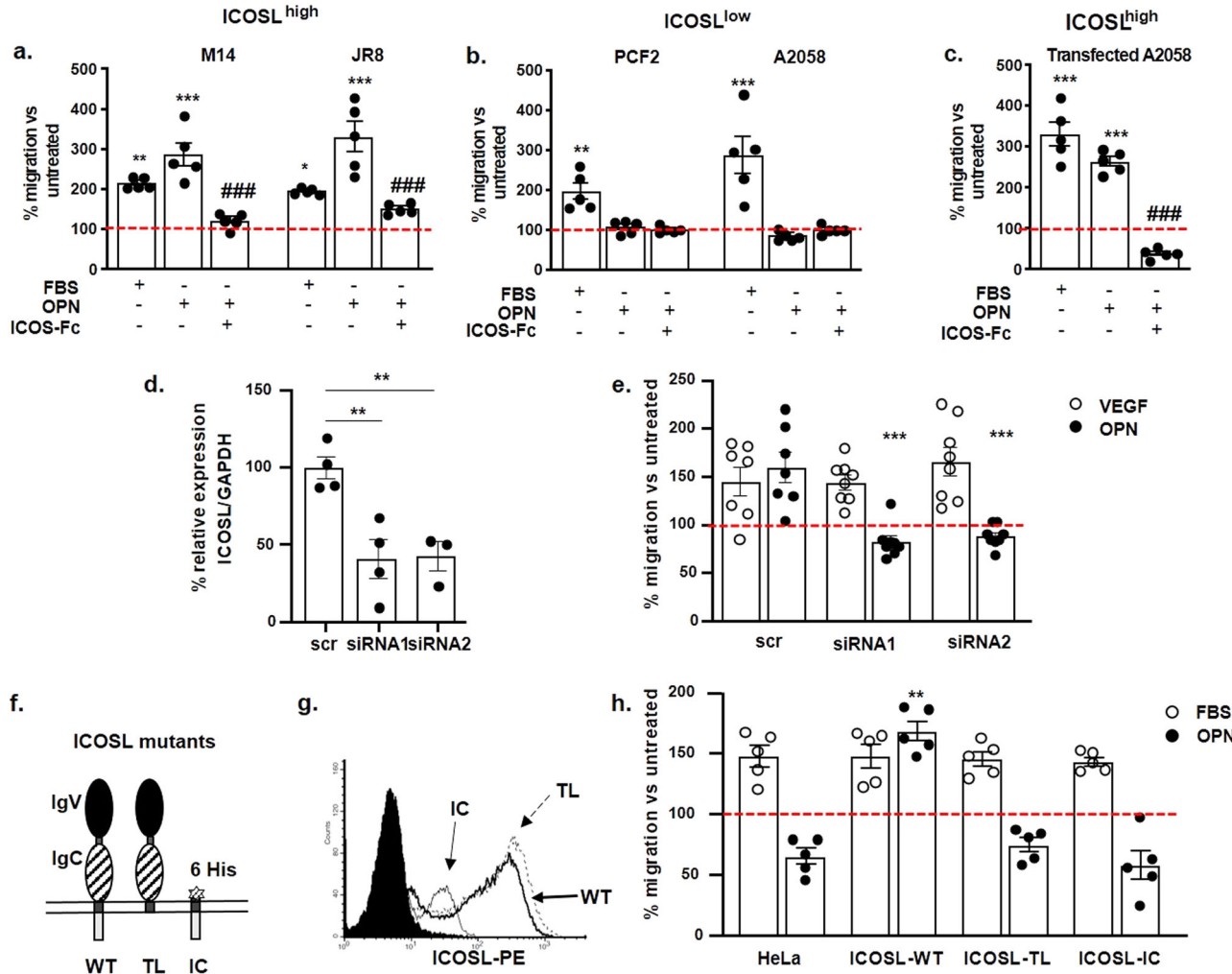

**Fig. 1 Expression of ICOSL is required for OPN-mediated cell migration and is blocked by ICOS-Fc. a** M14 and JR8 cell lines with high expression of ICOSL (ICOSL^high), **b** PCF2 and A2058 cell lines with low ICOSL expression (ICOSL^low), and **c** A2058 ICOSL^high-transfected cells were treated or not with ICOS-Fc (2 μg ml⁻¹) and then plated onto the apical side of Matrigel-coated filters in 50 μl serum-free medium. The cells were allowed to migrate for 8 h to the lower chamber containing 10 μg ml⁻¹ OPN or 20% FBS as chemotactic stimuli. The cells that migrated to the bottom of the filters were stained using crystal violet and all counted (quadruplicate filter) using an inverted microscope. Dotted line refers to untreated cells (NT). Data are expressed as the number of migrated cells per high-power field (***P < 0.001 refers baseline, ##P < 0.001 refers to OPN-induced migration (n = 5 technical replicate). **d** Relative quantification of ICOSL gene in silenced HUVEC using siRNA1 and siRNA2; cells treated with non-specific siRNA (SCR-siRNAscr) served as controls. Data are shown as gene expression relative to the expression of the endogenous control GAPDH (2^−ΔCt method) and T test was used; **P < 0.01 (n = 3–4 technical replicate). **e** Assessment of migration of silenced HUVECs, via a Boyden chamber assay, in response to OPN and VEGF (***P < 0.001 refers to OPN-mediated migration in scr sample) (n = 7–8 technical replicate). **f** Schematic overview of the three different mutants of ICOSL generated. **g** ICOSL expression on transfected HeLa cells as assessed by flow cytometry. In the dot plots, gray, gray dotted, and black lines indicate the intracytoplasmic mutant (IC), the tailless mutant (TL), and the wild-type molecule (WT), respectively. The black filled histogram shows the negative control. **h** HeLa migration induced by OPN or FBS was assessed via a Boyden chamber assay. Cells migration was evaluated as described in **a**, **b**, and **c** (**P < 0.01 refers to baseline) (n = 5 technical replicate). All data are expressed as means ± standard error. For migration experiments one-way ANOVA with post-hoc Tukey multi-comparison test was used.

and open the possibility of a direct binding between the two molecules. To test this hypothesis, an ELISA-based interaction assay was run, using recombinant OPN as the capture protein, and evaluating the binding of titrated amounts of soluble recombinant ICOSL-Fc. A commercial OPN produced in mammalian cells was used to ensure the presence of the physiological post-translational modifications. Results showed that ICOSL displays concentration-dependent binding to OPN (Fig. 2a). The same experiments were also performed using a 1:1 mixture of ICOSL-Fc and ICOS-Fc to assess any competition between OPN and ICOS-Fc for ICOSL binding or using ICOS-Fc alone to rule out the interaction with the Fc domain. Results showed that the

presence of ICOS-Fc did not inhibit ICOSL-Fc binding to OPN, and ICOS-Fc alone did not bind to OPN (Fig. 2a). Since the commercial OPN corresponds to OPN-b, a splicing variant missing the exon 5, the full-length OPN (OPN-a) was produced in an eukaryotic system and its binding capability to ICOSL-Fc tested by ELISA. Further, the role of OPN post-translational modifications in OPN/ICOSL binding was assessed, using a recombinant OPN (OPN-GST) produced in *E. Coli*. As shown, with both preparations, the results were similar to those obtained using the commercial OPN-b (Supplementary Fig. 2), indicating that OPN/ICOSL binding does not require post-translational modifications. To strengthen these results, reciprocal ELISA was

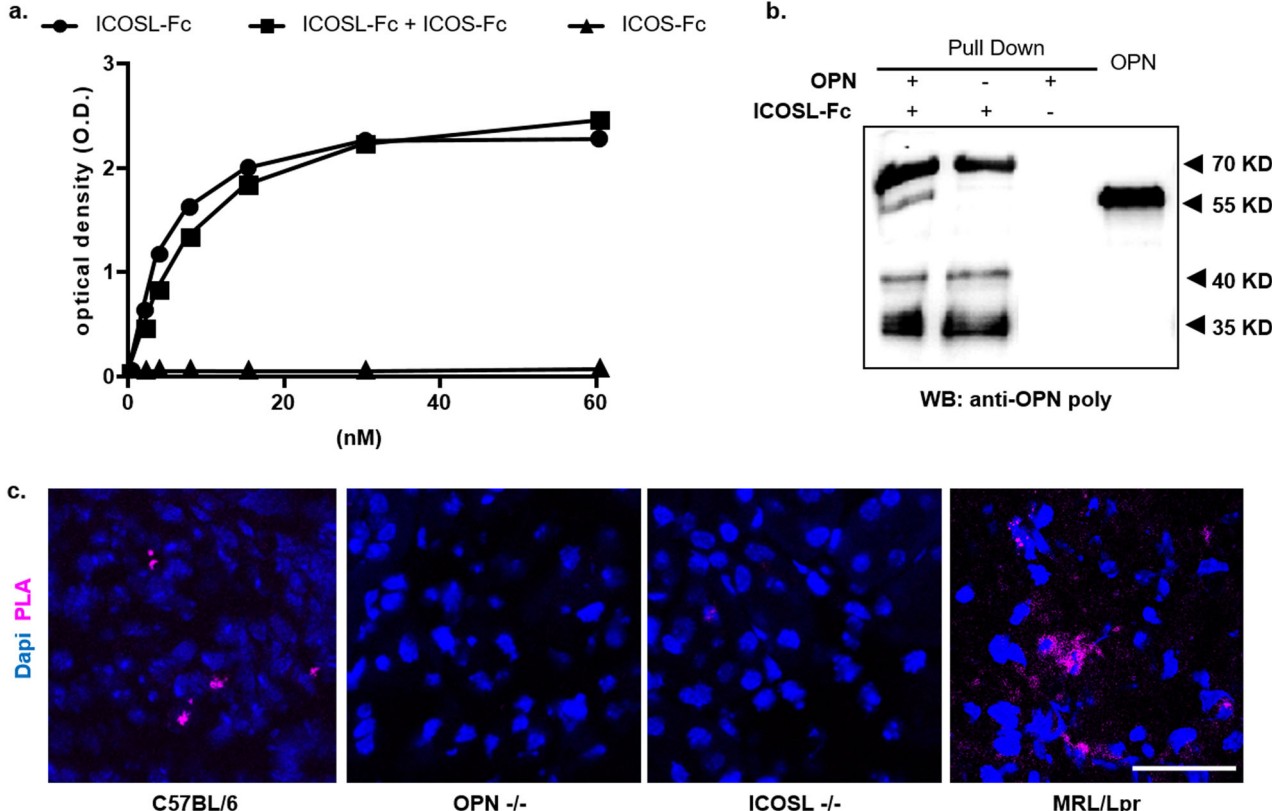

**Fig. 2 OPN binds the extracellular portion of ICOSL. a** ELISA-based interaction assay. Recombinant OPN was used as capture protein on 96 well plates and the binding of titrated amounts of soluble recombinant ICOSL-Fc was evaluated alone (black circle) or mixed 1:1; with ICOS-Fc (black square) or of ICOS-Fc alone (black triangle). Data are expressed as means ± standard error ($n = 3$ technical replicate). **b** Pull-down assay. Schematic presentation of the pull-down assay used to detect ICOSL/OPN binding, in which ICOSL-Fc was used as a Sepharose-bound bait protein incubated (first lane) with OPN. The second (ICOSL-Fc) and third lanes (OPN) represent the negative controls in which only one of the two interactors were present. The membrane was blotted with an anti-OPN polyclonal antibody. A representative immunoblot of two independent experiment is shown (uncropped blot is shown in Supplementary Fig. 6). **c** Proximity Ligation Assay (PLA) in mice kidneys. PLA signal (violet) shows protein-protein interactions between ICOSL and OPN; C57BL/6J and MRL/Lpr chosen as positive controls, while OPN-/- and ICOSL-/- as negative ones. Four images at a magnification of ×40 were analyzed for each sample, considering 3 mice per group ($n = 3$ biological replicate).

performed by coating the plate with ICOSL-Fc and testing binding of titrated amounts of commercial OPN-b. In this case, too, OPN-b displays concentration-dependent binding to ICOSL (Supplementary Fig. 3).

To further confirm the OPN/ICOSL interaction, pull-down experiments were run with recombinant OPN-b and ICOSL-Fc. Precipitated proteins were detected with an anti-OPN antibody. Results showed that OPN co-immunoprecipitated with ICOSL-Fc, confirming the ELISA results (Fig. 2b).

Collectively, these data suggest that: (i) the extracellular portion of ICOSL binds directly to OPN, (ii) this interaction occurs at a different binding site compared to ICOS-Fc, (iii) OPN binding to ICOSL is independent of its post-translational modifications.

To evaluate the OPN/ICOSL interaction in a tissue setting, a proximity ligation assay (PLA) was run that enables binding between proteins that reside at <40 nm distance in tissue sections to be detected. PLA assay on kidney sections (expressing both OPN and ICOSL) showed sparse distinct spots of interaction in wild-type C57BL/6 mice and multiple diffuse spots of interaction in MRL-lpr mice, (Fig. 2c). MRL-lpr mice, spontaneously develop glomerulonephritis, increasing expression level of both proteins and have been chosen as a model hyper-expressing both molecules[20,21]. By contrast, no PLA signal was detected in OPN−/− and ICOSL−/− mice, chosen as negative controls.

**Conformational analysis of the OPN/ICOSL complex by limited proteolysis.** Limited proteolysis and chemical cross-linking experiments were employed for conformational studies, to map the interaction regions of OPN and ICOSL. We optimized conditions, to aim enzymatic activity only towards the most exposed and flexible regions, so that regions of OPN and ICOSL, exposed to proteolysis in the isolated proteins and hidden in the complex, may be identified. These may be part of the protein-protein interaction region. Since OPN was more susceptible to hydrolysis than ICOSL-Fc with an enzyme vs. substrate ratio (E:S) 50 times lower for OPN than for ICOSL-Fc, using either trypsin or chymotrypsin. These data suggest OPN has either greater flexibility or a less compact structure than ICOSL-Fc, in line with the intrinsically disordered nature of OPN[16]. The experiments were thus performed on the complex using the lower E:S ratio, thus giving priority to the conformational analysis of OPN rather than ICOSL-Fc, whose digestion would require more protease. As shown in Fig. 3, some differences among the isolated proteins and the digestion profiles of the complex were highlighted. As expected, in both tryptic and chymotryptic hydrolyses, a large amount of ICOSL-Fc remained undigested, without substantial changes, also in the material digested from the complex. By contrast, the proteolytic profiles of isolated OPN displayed low-molecular-weight bands (Figs. 3a-1 to 5) that are absent upon

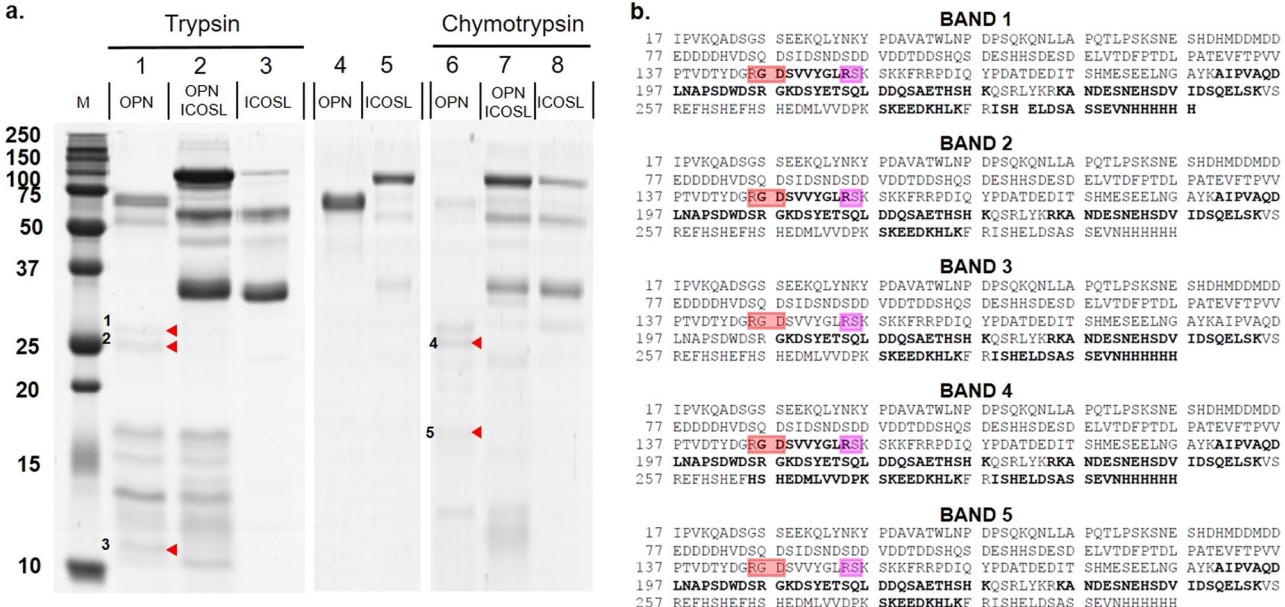

**Fig. 3 Conformational analysis of the OPN/ICOSL complex by limited proteolysis. a** SDS-PAGE of proteolytic mixture of OPN, ICOSL-Fc, and their complex, after 30 min of proteolysis with trypsin (left panel) and chymotrypsin (right panel). Lane M: molecular weight markers; lane 1: OPN (5 μg) digested with trypsin, E:S 1:5000; lane 2: OPN/ICOSL complex digested with trypsin, E:S 1:5000; lane 3: ICOSL (5 μg) digested with trypsin, E:S 1:100; lane 4: OPN (2 μg); lane 5: ICOSL (2 μg); lane 6: OPN (5 μg) digested with chymotrypsin, E:S 1:5000; lane 7: OPN–ICOSL 2:1 complex digested with chymotrypsin, E:S 1:5000; lane 8: ICOSL (5 μg) digested with chymotrypsin, E:S 1:100. **b** Sequence coverages obtained by tryptic mass mapping experiments performed on bands 1, 2, 3, 4, and 5 which were discriminant among isolated proteins and complex. In bold the regions mapped by LC-MS/MS analyses. RGD is boxed in red and Thrombin cleavage site in pink.

digestion of the OPN/ICOSL-Fc complex, suggesting that the cleavage sites that generate them are located in regions that are exposed in isolated OPN but buried upon interaction with ICOSL (Fig. 3a). These low-molecular-weight bands of OPN proteolysis were analyzed using a mass mapping approach. Results showed that the bands with electrophoretic mobility of about 25–26 kDa (bands 1, 2, and 4) mapped in the middle of OPN, upstream of position 130, suggesting that residues close to this amino acid residue are exposed and accessible to proteases in the isolated protein, but become hidden in the OPN/ICOSL-Fc complex. Bands 3 and 5 mapped in the same OPN C-terminal portion as bands 1, 2, and 4, but they covered a substantially shorter sequence, which suggests that they were generated by sub-digestion events (Fig. 3b). These data suggest that the region of OPN interaction with ICOSL-Fc is located in the central part of the protein near to the RGD motif and the physiological thrombin cleavage site. These experiments gave no hint concerning the ICOSL-Fc, since it was substantially undigested at the E:S ratio used. This is a common and unavoidable effect of operating with two proteins having such different susceptibilities to proteases.

**Conformational analysis of the OPN/ICOSL complex by cross-linking.** Additional experiments were carried out using the bifunctional cross-linker 3,3′dithiobis(sulfosuccinimidyl propionate) (DTSSP). DTSSP is a lysine–lysine specific reactive with a spacer arm 12 Å in length that can be cleaved under reducing conditions. Cross-linkers can also provide information on the tertiary structure since they may be considered as actual chemical labels: the presence/absence of DTSSP-mediated modification of specific lysines provides an indication of their exposure within isolated proteins or the complex. An excess of DTSSP was added to the pre-formed complex between GST-OPN and ICOSL. As control, the same treatment was applied to isolated GST-OPN and ICOSL. Then, the samples were separated by non-reducing

naïve-PAGE and stained by colloidal Coomassie (Fig. 4a). Lane 2, where the complex was loaded after reaction with DTSSP, shows the appearance of a 100 kDa band (marked with a bracket) with an electrophoretic mobility higher than those corresponding to each isolated protein (marked with arrows), as expected for the GST-OPN/ICOSL complex. This band was cut out and in situ digested with trypsin, in non-reducing conditions to preserve the integrity of the cross-linker moiety. LC-MS/MS and MALDI-MS analyses of the peptide mixture detected the presence of peptides derived from both OPN and ICOSL, which shows that the band was generated by DTSSP-mediated covalent linking of GST-OPN and ICOSL and confirms direct interaction between the two molecules. However, the mass spectrometer data analysis did not provide direct and conclusive identification of the OPN–ICOSL cross-linking sites. Detecting actual cross-linked peptides within digested mixtures has long been problematic, because some cross-linked peptides are too large to detect, their fragmentation is more complicated than usual linear peptides, and not every protein or peptide is cross-linked in the same way[22,23].

To gain insight into the OPN/ICOSL surface topology[24], an aliquot of the peptide sample produced by digestion of the band containing the cross-linked proteins was reduced and analyzed by MALDI-MS and LC-MS/MS. Overall comparison of MS data from oxidized and reduced samples was informative concerning lysine exposure in OPN and ICOSL, whether analyzed as isolated molecules or their complex (Tables 1, 2).

With regard to ICOSL, all expected peptides containing a lysine residue were identified, in both isolated ICOSL and the complex, except for $K_{174}$. This residue was recognized by trypsin in isolated and untreated ICOSL, generating the peptide 175–194; once modified, the residue escaped trypsin hydrolysis, and occurred within an excessively large tryptic peptide that would be difficult to detect. In conclusion, failure to detect the peptide containing $K_{174}$ in the complex following DTSSP treatment suggests that the protein regions including this residue might be involved in OPN

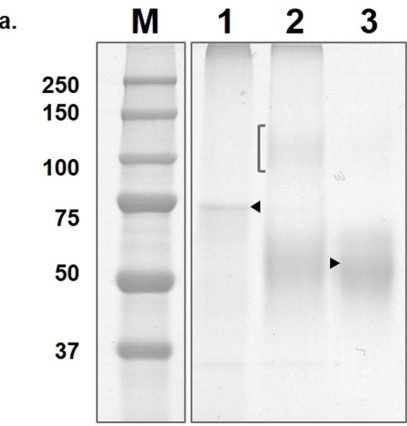

### b. ISOLATED ICOSL + DTSSP

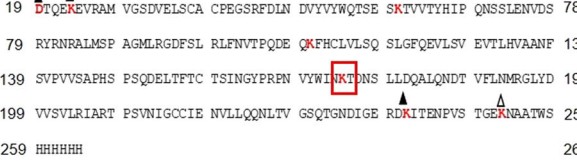

### c. ISOLATED OPN + DTSSP

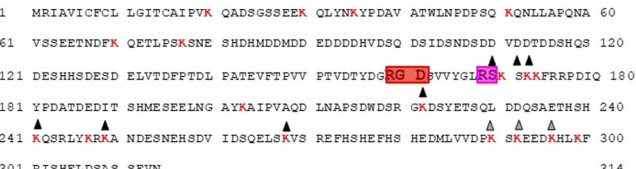

**Fig. 4 Conformational analysis of the OPN/ICOSL complex by cross-linking experiments. a** Non-reducing SDS-PAGE of DTSSP treated samples. Lane 1: GST-OPN; lane 2: complex GST-OPN–ICOSL; lane 3: ICOSL. The bracket indicates the presence of a faint band, migrating as per the electrophoretic mobility expected for the complex GST-OPN–ICOSL. ICOSL **b** and OPN **c** sequences with all lysine residues indicated in red, and lysines modified with squares and triangles, in both the isolated proteins (upper panels) and the complex (lower panels). Filled squares indicate lysines modified as dead-end by DTSSP; empty squares indicate those modified as dead-ends or intra-molecule cross-linking. Black triangles on OPN/ICOSL complex indicate lysines found modified in the complex, before and after the REDCAM reaction. Gray triangles represent lysines found modified only in the complex after the REDCAM reaction. Empty triangles are lysines found modified only before the REDCAM reaction. RGD is boxed in green and thrombin cleavage site in orange.

binding or in conformational changes (Fig. 4b). With regard to OPN, in both the complex and the isolated GST-OPN, the lysines $K_{222}$, $K_{241}$, $K_{249}$, and $K_{268}$ were labeled with a single dead-end molecule of DTSSP in all cases. Since these residues are equally exposed in both isolated and complexed GST-OPN, the possibility that they are located in regions involved in the interaction with ICOSL may be ruled out. Moreover, peptides 221–244 and 249–271 displayed a mass increased of 174 Da (accounting for the introduction of a DTSSP moiety cross-linking two lysines within the linear peptides), suggesting the occurrence of intramolecular cross-links between $K_{222}$ and $K_{241}$ and between $K_{249}$ and $K_{268}$. Therefore, the two pairs of cross-linked lysines are spatially close, and their side chains are oriented appropriately to be cross-linked.

$K_{247}$ was always detected in peptides including $K_{249,}$ both in the isolated protein (242–268 and 245–268 + 1CL$_{H2O}$) and in the complex (245–268 + 1CL$_{H2O}$); although a certain assignment of its state is not allowed, it is plausible to suppose it behaves similarly to $K_{241}$ and $K_{249}$ given its nearby location (Fig. 4b).

Residues $K_{20}$, $K_{30}$, $K_{35}$, $K_{51}$, $K_{70}$ $K_{77}$, and $K_{203}$ were in no case labeled by DTSSP within the complex, nor were they generated by

a cleavage event in correspondence with a non-modified lysine. These findings suggest that, in the GST-OPN/ICOSL complex as well as in OPN alone, none of these residues are ever exposed to be modified by the cross-linking agent.

Intriguingly, $K_{170}$, $K_{172}$, and $K_{173}$ were modified by DTSSP only when OPN was bound to ICOSL, suggesting that the region including these three residues undergoes a significant conformational change upon complex formation, exposing residues whose side chains are concealed or are involved in local interactions in the isolated protein. Noteworthy, these three lysine residues are located next to the thrombin cleavage site ($R_{168}$–$S_{169}$) and are thus downstream of both the RGD motif and the potential interaction region with ICOSL, as suggested by the limited proteolysis experiments. When the cross-linked complex was reduced and alkylated, the peptide 272–299 was identified, suggesting that $K_{299}$ may not be modified by DTSSP. Moreover, this peptide, which contains three other lysine residues ($K_{290}$, $K_{292}$, and $K_{296}$), was found to be doubly modified: upon both reduction and alkylation reactions. This finding suggests that two of these three lysine residues are labeled within the complex. The reduced and alkylated forms of this peptide might be generated

**Table 1 List of peptides identified by the mass mapping procedure and relative to: isolated OPN, isolated OPN + DTSSP (indicated with CL); OPN/ICOSL complex + DTSSP (indicated with CL); OPN/ICOSL + DTSSP (indicated with CL) following reduction and alkylation treatment (indicated with CAM).**

| Lysine Residue | Peptides from OPN | Peptides from (OPN + DTSSP) | Peptides from the complex (OPN/ICOSL + DTSSP) | Peptides from the complex (OPN/ICOSL + DTSSP) REDCAM |
|---|---|---|---|---|
| **K20** | – | – | – | – |
| **K30** | – | – | 21–35 Exp 1682.84 Theor 1682.77 | – |
| **K35** | 31–51 Exp 2447.01 Theor 2447.21 | 31–51 (+1CL$_{H2O}$) Exp 2639.19 Theor 2639.21 | 21–35 Exp 1682.84 Theor 1682.77 | – |
| **K51** | 36–51 Exp 1800.78 Theor 1800.87 | 36–51 Exp 1800.80 Theor 1800.87 | – | – |
| **K70** | 52–77 Exp 2887.33 Theor 2877.42 | 52–77 Exp 2887.37 Theor 2877.42 | 52–77 Exp 2887.31 Theor 2877.42 | – |
| **K77** | 52–77 Exp 2887.33 Theor 2877.42 | 52–77 Exp 2887.37 Theor 2877.42 | 52–77 Exp 2887.31 Theor 2877.42 | – |
| **K170** | – | 160–172 Exp 1394.60 Theor 1394.75 | 169–175 (+1CL$_{H2O}$ + 1CL$_{INTRA}$) Exp 1245.46 Theor 1245.53 | 169–175 (+3CL + 3CAM) Exp 1314.54 Theor 1314.53 |
| **K172** | – | 160–172 Exp 1394.60 Theor 1394.75 | 169–175 (+1CL$_{H2O}$ + 1CL$_{INTRA}$) Exp 1246.46 Theor 1245.53 | 169–175 (+3CL + 3CAM) Exp 1314.54 Theor 1314.53 |
| **K173** | – | – | 169–175 (+1CL$_{H2O}$ + 1CL$_{INTRA}$) Exp 1246.46 Theor 1245.53 | 169–175 (+3CL + 3CAM) Exp 1314.54 Theor 1314.53 |
| **K203** | 176–203 Exp 3223.33 Theor 3223.42 | 176–203 Exp 3223.38 Theor 3223.42 | 176–203 Exp 3223.25 Theor 3223.42 | – |
| **K222** | – | 221–244 (+1CL$_{H2O}$) Exp 2925.22 Theor 2925.22; 221–244 (+1CL$_{INTRA}$) Exp 2907.24 Theor 2907.22 | 221–244 (+1CL$_{H2O}$) Exp 2925.13 Theor 2925.22; 221–244 (+1CL$_{INTRA}$) Exp 2907.14 Theor 2907.22 | 221–244 (+1CL + 1CAM) Exp 2878.16 Theor 2878.22; 221–244 (+2CL + 2CAM) Exp 3023.18 Theor 3023.22 |
| **K241** | 223–241 Exp 2176.82 Theor 2176.91 | 223–244 (+1CL$_{H2O}$) Exp 2740.08 Theor 2740.11; 221–244 (+1CL$_{H2O}$) Exp 2925.22 Theor 2925.22 | 221–244 (+1CL$_{H2O}$) Exp 2925.13 Theor 2925.22; 221–244 (+1CL$_{INTRA}$) Exp 2907.14 Theor 2907.22 | 223–244 (+1CL + 1CAM) Exp 2693.04 Theor 2693.11; 221–244 (+1CL + 1CAM) Exp 2878.17 Theor 2878.22; 221–244 (+2CL + 2CAM) Exp 3023 .18 Theor 3023.22 |
| **K247** | 242–268 Exp 3175.62 Theor 3175.55 | 242–268 Exp 3175.57 Theor 3175.55; 245–268 (+1CL$_{H2O}$) Exp 2996.10 Theor 2996.36 | 245–268 (+1CL$_{H2O}$) Exp 2996.27 Theor 2996.36 | – |
| **K249** | 249–268 Exp 2243.92 Theor 2244.01 | 242–268 Exp 3175.57 Theor 3175.55; 249–268 (+1CL$_{H2O}$) Exp 2435.99 Theor 2436.01; 245–268 (+1CL$_{H2O}$) Exp 2996.10 Theor 2996.36; 249–271 (+1CL$_{INTRA}$) Exp 2760.20 Theor 2760.22 | 249–268 (+1CL$_{H2O}$) Exp 2435.95 Theor 2436.01; 245–268 (+1CL$_{H2O}$) Exp 2996.27 Theor 2996.36 | 249–268 (+1CL + 1CAM) Exp 2388.94 Theor 2389.01 |
| **K268** | 249–268 Exp 2243.92 Theor 2244.01 | 249–271 (+1CL$_{H2O}$) Exp 2778.19 Theor 2778.22; 249–271 (+2CL$_{H2O}$) Exp 2970.22 Theor 2970.22; 249–271 (+1CL$_{INTRA}$) Exp 2760.20 Theor 2760.22 | 249–271 (+1CL$_{H2O}$) Exp 2778.19 Theor 2778.22; 249–271 (+2CL$_{H2O}$) Exp 2970.12 Theor 2970.22; 249–271 (+1CL$_{INTRA}$) Exp 2760.12 Theor 2760.22 | 249–271 (+2CL + 2CAM) Exp 2876.17 Theor 2876.22 |
| **K290** | 272–292 Exp 2534.01 Theor 2534.16 | – | – | 272–299 (+2CL + 2CAM) Exp 3703.41 Theor 3703.61 |
| **K292** | 272–292 Exp 2534.01 Theor 2534.16 | – | – | 272–299 (+2CL + 2CAM) Exp 3703.41 Theor 3703.61 |

**Table 1 (continued)**

| Lysine Residue | Peptides from OPN | Peptides from (OPN + DTSSP) | Peptides from the complex (OPN/ICOSL + DTSSP) | Peptides from the complex (OPN/ICOSL + DTSSP) REDCAM |
|---|---|---|---|---|
| K296 | – | – | – | 272–299 (+2CL + 2CAM) Exp 3703.41 Theor 3703.61 |
| K299 | – | – | – | 272–299 (+2CL + 2CAM) Exp 3703.41 Theor 3703.61 |

Lysine residue positions within the OPN sequence are shown together with the identified peptides, their molecular weights expressed as monoisotopic, and the experimental molecular weight obtained by MALDIMS or nanoLC-MS/MS analyses.

either from inter- or from intramolecular cross-links, not ruling out the possible involvement of this C- terminal region in ICOSL binding (Fig. 4c).

**ICOSL binds both the N-term and the C-term thrombin-generated fragments of OPN.** Collectively these data suggest that, on ICOSL, the OPN binding site is located in the IgC domain (Fig. 5a), whereas within OPN, two different binding sites for ICOSL exist. One is upstream of the RGD domain, and the other on the C-terminal portion of the protein (Fig. 5b). Since the thrombin cleavage site in OPN is in the middle of these putative sequences, the ability to bind ICOSL displayed by recombinant forms of the N-term (aa 17–167) and C-term (aa 168–314) portions of OPN was tested; these correspond to the two fragments generated by the physiologic thrombin cleavage of the whole OPN (Fig. 6a)[12]. ELISA assay showed that both the OPN portions bind to ICOSL, but to a lesser extent than the full-length protein (Fig. 6b). This supports the existence of multiple ICOSL binding sites in OPN.

**ICOSL expression promotes tumor-cell migration while inhibiting anchorage-independent cell growth.** To better investigate the effects of the OPN/ICOSL interaction on tumor progression and metastatization, we employed the orthotopic 4T1 mammary carcinoma which spontaneously metastasizes to the lungs. Since 4T1 cells secrete high levels of OPN, but do not express ICOSL, the cells were stably transfected with a plasmid encoding ICOSL, to generate 4T1 cells expressing high levels of ICOSL (4T1$^{ICOSL}$). Intriguingly, 4T1$^{ICOSL}$ cells expressed more OPN than the parent cells, suggesting that ICOSL may upregulate OPN in these cells (Supplementary Fig. 4a). In line with previous findings, migration assays showed that OPN induces migration of 4T1$^{ICOSL}$ cells but not of 4T1 and it was inhibited by ICOS-Fc (Fig. 7a).

To compare the neoplastic potential of 4T1 and 4T1$^{ICOSL}$ cells, their growth kinetic in adhesion-dependent and -independent conditions was evaluated, in the presence and absence of OPN. Adhesion-dependent growth, assessed by growing cells in standard culture plates, showed that 4T1 and 4T1$^{ICOSL}$ cells display the same growth kinetic, even in the presence of OPN (Supplementary Fig. 4b). Conversely, adhesion-independent growth, assessed by culturing cells in soft agar, showed that 4T1$^{ICOSL}$ cells grow less strongly than 4T1 cells. Interestingly, addition of OPN further inhibited the growth of 4T1$^{ICOSL}$ cells in a dose-dependent manner but left that of 4T1 cells unchanged (Fig. 7b).

These results suggest that expression of ICOSL decreases the growth potential of 4T1 cells but increases their migratory activity, and that these effects depend on OPN.

**Tumor growth, angiogenesis and dissemination are influenced by OPN/ICOSL binding.** To assess the effect of the OPN/ICOSL interaction in tumor, parent 4T1 or 4T1$^{ICOSL}$ were injected into syngeneic mice (Fig. 8a). We observed that, even though the tumor volume was significantly smaller in 4T1$^{ICOSL}$ tumors as compared with 4T1 (grown and measured in parallel- Fig. 8b) the number of lung metastasis was similar in both the tumor types (Fig. 8c). No effect was observed on mice body weight (Supplementary Fig. 5). These results are in line with the initial hypothesis of the pro-invasive role of the OPN/ICOSL interaction in the tumor context.

We checked OPN and ICOSL expression within the tumor by immunofluorescence and we found that 4T1$^{ICOSL}$ tumors expressed high levels of both OPN and ICOSL and displayed a high degree of colocalization. By contrast, in 4T1 tumors ICOSL expression was very low, possibly ascribable to stromal cells and infiltrating immune cells (Fig. 8d). PLA analysis revealed high signals in 4T1$^{ICOSL}$ tumors; notably these signals were more abundant, aggregated and organized along a linear front in the tumor periphery. Conversely, few signals were detected in 4T1 tumors (Fig. 8e).

The impact of the OPN/ICOSL interaction on the tumor microenvironment was then investigated, focusing on angiogenesis, which favors tumor dissemination. Confocal analysis showed that 4T1$^{ICOSL}$ tumors displayed higher vessel density than 4T1 tumors (Fig. 8f).

## Discussion

This study shows that ICOSL is a hitherto-undescribed receptor for OPN and that interaction between the two is involved in promoting endothelial and tumor-cell migrations. Moreover, although the OPN/ICOSL interaction may restrain primary tumor growth both in vitro and in vivo, it seems to favor its potential metastatic dissemination, increasing tumor angiogenesis.

The direct interaction between OPN and ICOSL is shown by their bidirectional ability to bind to one another, as detected by ELISA-based assays and confirmed by pull-down experiments. These experiments also show that ICOS and OPN bind different sites of ICOSL, since they do not compete for binding. This model was confirmed by the chemical cross-linking experiments, suggesting that the OPN binding site on ICOSL involves a region around $K_{174}$ which is located in the membrane proximal IgC domain of ICOSL, whereas ICOS is known to bind the membrane distal to the IgV domain.

With regard to OPN, the limited proteolysis approach maps the ICOSL binding site in the central part of OPN, upstream of the RGD motif, and the thrombin cleavage site. Furthermore, cross-linking experiments indicate that the region downstream the RGD domain (including $K_{170}$, $K_{172}$, and $K_{173}$) becomes exposed upon ICOSL binding, which in turn indicates that, although OPN is known to behave as an IDP in solution[16], it may acquire a specific conformation in the complex with ICOSL. This conformational change might influence accessibility of the nearby

**Table 2 Peptides identified by the mass mapping procedure and relative to: isolated ICOSL, isolated ICOSL + DTSSP (indicated with CL); OPN/ICOSL complex +DTSSP (indicated with CL); OPN/ICOSL + DTSSP (indicated with CL) following reduction and alkylation treatment (indicated with CAM).**

| Lysine Residue | Peptides from ICOSL | Peptides from (ICOSL + DTSSP) | Peptides from the complex (OPN/ICOSL + DTSSP) | Peptides from the complex (OPN/ICOSL + DTSSP) REDCAM |
|---|---|---|---|---|
| **N-TERM** | 19–23<br>Exp 619.24<br>Theor 619.28<br>19–26<br>Exp 1003.47<br>Theor 1003.49 | 19–26(+1CL$_{INTRA}$)<br>Exp 1177.44<br>Theor 1177.49<br>19–26(+1CL$_{H2O}$)<br>Exp 1195.45<br>Theor 1195.49 | 19–26(+1CL$_{INTRA}$)<br>Exp 1177.44<br>Theor 1177.49<br>19–26(+1CL$_{H2O}$)<br>Exp 1195.45<br>Theor 1195.49 | 19–26(+1CL + 1CAM)<br>Exp 1148.47<br>Theor 1148.49<br>19–26(+2CL + 2CAM)<br>Exp 1293.47<br>Theor 1293.49 |
| **K23** | 19–26<br>Exp 1003.47<br>Theor 1003.49 | 19–26(+1CL$_{INTRA}$)<br>Exp 1177.44<br>Theor 1177.49<br>19–26(+1CL$_{H2O}$)<br>Exp 1195.45<br>Theor 1195.49 | 19–26(+1CL$_{INTRA}$)<br>Exp 1177.44<br>Theor 1177.49<br>19–26(+1CL$_{H2O}$)<br>Exp 1195.45<br>Theor 1195.49 | 19–26(+1CL + 1CAM)<br>Exp 1148.47<br>Theor 1148.49<br>19–26(+2CL + 2CAM)<br>Exp 1293.47<br>Theor 1293.49 |
| **K60** | 45–60<br>Exp 1992.86<br>Theor 1992.91 | 45–60<br>Exp 1992.86<br>Theor 1992.91<br>61–79<br>Exp 2158.01<br>Theor 2158.06 | 45–60<br>Exp 1992.83<br>Theor 1992.91<br>61–79<br>Exp 2158.00<br>Theor 2158.06 | – |
| **K110** | 100–110<br>Exp 1317.61<br>Theor 1317.66 | – | – | – |
| **K174** | 175–194<br>Exp 2307.13<br>Theor 2307.12 | 175–194<br>Exp 2307.99<br>Theor 2307.12 | – | – |
| **K241** | 240–252<br>Exp 1416.67<br>Theor 1416.71 | 240–252(+1CL$_{H2O}$)<br>Exp 1608.70<br>Theor 1608.71 | 240–252<br>Exp 1416.64<br>Theor 1416.71<br>240–252(+1CL$_{H2O}$)<br>Exp 1609.64<br>Theor 1609.71 | 240–252 (+1CL + 1CAM)<br>Exp 1562.66<br>Theor 1562.71 |
| **K252** | 242–264<br>Exp 2626.16<br>Theor 2626.22 | 242–264 (+1CL$_{H2O}$)<br>Exp 2818.18<br>Theor 2818.22<br>253–264<br>Exp 1470.61<br>Theor 1470.64 | 242–264(+1CL$_{H2O}$)<br>Exp 2818.18<br>Theor 2818.22<br>253–264<br>Exp 1470.61<br>Theor 1470.64 | 253–264<br>Exp 1470.61<br>Theor 1470.64 |

Lysine residues position within the OPN sequence are reported together with the identified peptides, their molecular weights expressed as as monoisotopic, and the experimental molecular weight obtained by MALDIMS or nano LC-MS/MS analyses.

thrombin ($R_{168}S_{169}$) and ($S_{162}LAYGLR_{168}$) MMP3/7 cleavage sites. This may be biologically relevant, since the full-length OPN, and each fragment generated by its enzymatic processing in inflammatory conditions, have partly distinct activities[12]. For instance, full-length OPN induces both cell adhesion and migration, whereas the N-terminal and C-terminal fragments have selective effects on cell adhesion and cell migration, respectively. Moreover, transglutaminase-cross-linked OPN, which forms proteolysis-resistant inactive OPN polymers, reduces breast cancer cell invasion and migration in vitro[25]. Conversely, the OPN fragments generated by thrombin cleavage increase hepatocellular carcinoma cell invasion[26].

Since the putative interaction region, shown by the limited proteolysis experiments to be the OPN region located upstream of the thrombin cleavage site ($R_{168}S_{169}$), does not contain lysine residues, for the long stretch to $K_{77}$ the cross-linking experiments were uninformative for that region. However, they were informative concerning the possible existence of a second interaction site for ICOSL, comprising the OPN residues $K_{290}$, $K_{292}$, and $K_{296}$, located close to the OPN C-terminus. Experiments using recombinant forms of OPN, corresponding to the C- and N-terminal fragments generated by thrombin, confirmed that both

fragments can bind ICOSL, although to lesser extents than the full-length molecule. The finding that bacteria-produced OPN can bind to ICOSL indicates that eukaryotic post-translational modifications, such as phosphorylation and glycosylation[27], which are considered important in modulating some OPN functions, are not required for binding. The interaction between OPN and ICOSL was also confirmed, ex vivo, by PLA, showing that ICOSL and OPN interact in tissues that co-express them in close proximity, like in kidney sections; this is particularly evident in the chronic glomerulonephritis context spontaneously developed by MRLlpr/lpr mice where both molecules are expressed at high levels[20,21]. Their colocalization was detected by both immunofluorescence and PLA analysis also in 4T1[ICOSL] tumors. Notably, the two proteins are strongly expressed especially at the invasive front of the tumor, corroborating their involvement in cancer invasion. Indeed, despite the dimensions of the primary tumors, which are significantly smaller in 4T1[ICOSL] then in 4T1 mice, both groups display a similar number of lung metastasis. Based on the obtained results, our hypothesis is that OPN/ICOSL interaction may favor the metastatic activity of 4T1[ICOSL] tumors by enhancing both tumor-cell migration and angiogenesis. Indeed, OPN is a known proangiogenic factor, and this study

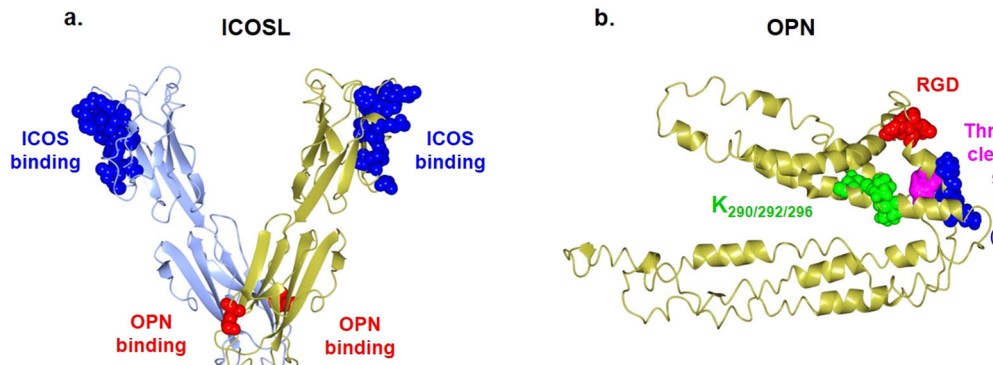

**Fig. 5 Conformational analysis of the OPN/ICOSL complex. a** Model for ICOSL created with the SWISS-MODEL server using 4f9p.1.A (CD277/Butyrophilin-3) as a template. Residues buried at the interface with ICOS are blue stained, K174 is red stained. **b** OPN is an intrinsically disordered protein that simultaneously exhibits extended, random coil-like conformations and stable, cooperatively-folded conformations[16]. OPN model from Dianzani et al., 2017[34]. RGD in red, $K_{170}$, $K_{172}$, and $K_{172}$ in blue, while $K_{290}$, $K_{292}$, and $K_{296}$ are in green. Thrombin cleavage site (RS) is shown in pink.

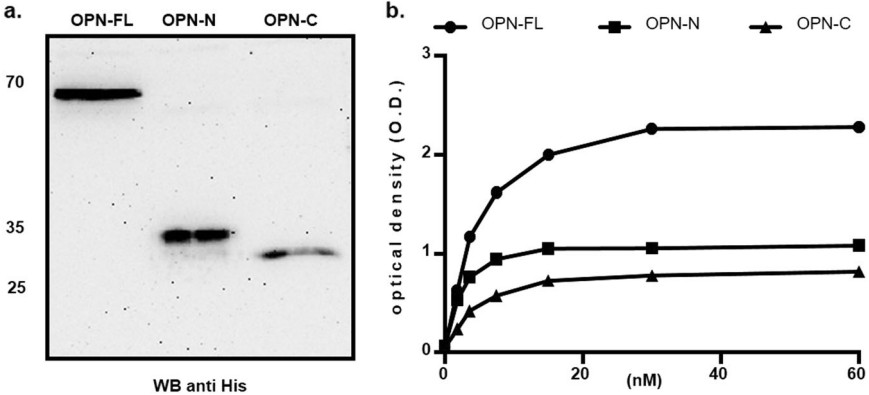

**Fig. 6 ICOSL binds both the N-term and the C-term thrombin-generated fragments of OPN. a** Western blot showing the OPN recombinant proteins (FL: full length, N and C-terminal fragments) probed with the anti-His-tag antibody (uncropped blot is shown Supplementary Fig. 6); **b** ELISA-based interaction assay. The graph shows the interaction of titrated amounts of soluble ICOSL-Fc with a fixed amount of OPN-FL, OPN N- or C-fragments coated on the plate. (black circle) shows OPN full length, (black square) OPN-N and (black triangle) OPN-C. Data are expressed as means ± standard error ($n = 3$ technical replicates).

shows that ICOSL expression in HUVECs is required for their migratory response to OPN in vitro.

The functional relevance of the OPN/ICOSL interaction in tumor-cell migration was highlighted by data showing that ICOSL expression in tumor cells is required for their migratory response to OPN. The functional effect of OPN on triggering cell migration needs the entire ICOSL molecule: it is lost when cells express truncated forms of ICOSL, lacking either the extracellular or the intracellular portion. However, attempts to identify this signaling directly by Western blot, by comparing wide phosphorylating events triggered by OPN in cells expressing or not expressing ICOSL, were inconclusive probably because of the multiple signaling pathways triggered by OPN, that interact contemporarily with its multiple receptors expressed on the same cells, comprising several integrins and CD44, beside ICOSL.

In this light, it is worth noting that the cross-linking experiments suggest that, upon its binding to ICOSL, the C-terminal portion of OPN, involving the residues $K_{170}$, $K_{172}$, and $K_{173}$, located close to the ICOSL binding site, undergoes a conformational change, which might influence the accessibility to integrin of the nearby RGD motif of OPN. The flexible and dynamic structure of OPN enables it to adopt different functional structures, i.e. folding upon binding, while permitting it to enter into multiple interactions with different binding partners[16]. It is, therefore, possible that OPN acts as a scaffold, that can interact contemporarily with several different cell-surface receptors, and its interaction with ICOSL might modulate this activity. This would be in line with the hypothesis that integrins co-operate with other receptors (e.g.: RTK, Tie2, and growth hormones receptors) on the cell surface[28]. However, the present findings indicate ICOSL to be an essential component of the signaling machinery recruited by soluble OPN, and it appears to be absolutely required for soluble OPN-induced motility even in the presence of integrins. This requirement might be due to the

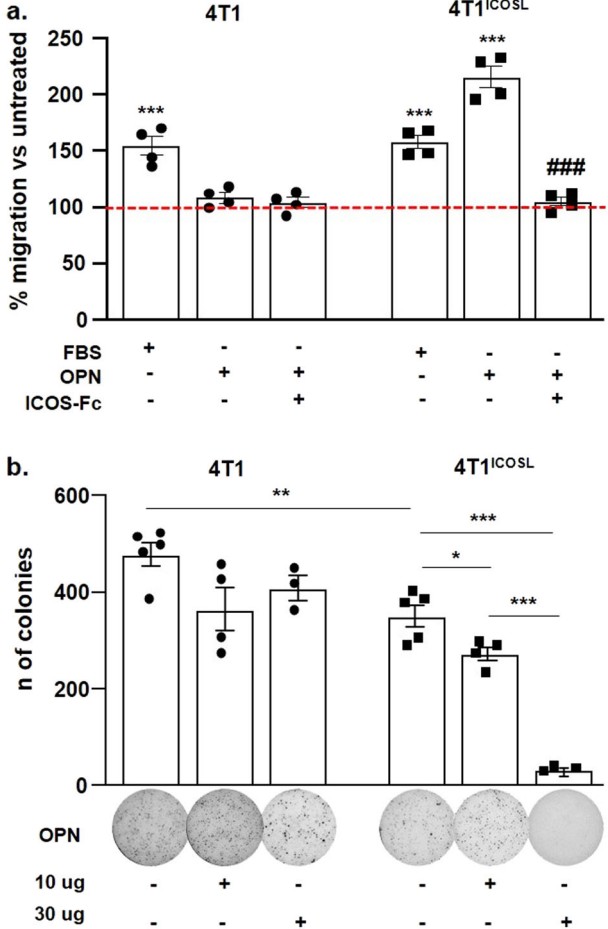

**Fig. 7 ICOSL expression promotes tumor-cell migration while inhibiting anchorage-independent cell growth. a** Migration assay of 4T1 and 4T1^ICOSL cells in response to OPN or ICOS-Fc, assessed via a Boyden chamber assay. 4T1 and 4T1^ICOSL cells were plated onto the apical side of Matrigel-coated filters in 50 μl serum-free medium in the presence or not of ICOS-Fc (2 μg ml⁻¹). OPN or FBS loaded in the basolateral chamber as chemotactic stimulus. The cells that migrated to the bottom of the filters were stained with crystal violet and all counted (quadruplicates filters) using an inverted microscope. The dotted line refers to untreated cells (NT). Data are expressed as the number of migrated cells per high-power field, ***$P <$ 0.001 refers to baseline, ###$P < 0.001$ refers to OPN-induced migration, one-way ANOVA with the post-hoc Tukey multi-comparison test was used ($n = 4$ technical replicate); **b** Anchorage-independent growth soft agar assay of 4T1 and 4T1^ICOSL transfected cells in response to OPN or PBS (NT). Colonies were counted using Image J Software. The images from one representative experiment are also shown. Please note that images are not fully self-explanatory of each histogram. For the comparison of 4T1 vs 4T1^ICOSL cells, the $T$ test was used. For the comparison of 4T1^ICOSL, 4T1^ICOSL + OPN one-way ANOVA with the post-hoc Tukey multi-comparison test was used (*$P < 0.05$, ** $P < 0.01$, *** $P < 0.001$). ($n = 3$-5 technical replicate). All data are expressed as means ± standard error.

mechano-sensing nature of integrins, which require traction for full activation, making soluble molecules poor activators[29].

The pro-migratory effect of OPN binding to ICOSL is dominantly antagonized by ICOS, since ICOS-Fc inhibits OPN-induced migration, even though this effect is not specific for OPN. However, ICOS-Fc can also display specific inhibitory effects on other OPN activities. On the one hand, ICOS-Fc inhibits ERK phosphorylation induced by OPN but not that

induced by ATP[8], while on the other hand, ICOS-Fc inhibits HUVEC tubulogenesis induced by OPN, but not that induced by VEGF-α[19].

An intriguing point is that, in 4T1^ICOSL cells, expression of ICOSL confers a dual response to OPN, which not only stimulates cell migration but also inhibits anchorage-independent growth in soft agar, whereas no effect is exerted on anchorage-dependent growth in cell culture wells. The different effect of OPN on the two types of growth might depend on the presence of strong integrin signals in anchorage-dependent growth, which may overcome negative effects mediated by OPN.

These results outline a scenario in which OPN/ICOSL interaction could play a dual role during tumor development by decreasing primary local tumor growth, and, in parallel, by mediating pro-invasive signals thus sustaining cancer cell dissemination. Although further experiments are needed to better clarify the role of OPN/ICOSL interaction in metastatic dissemination in vivo, this model would challenge the literature showing that primary 4T1 tumors disseminate cells to distant sites in function of the size of primary tumors, with large tumors producing more metastases than small ones[30]. Herein, we show that ICOSL overexpression seems to be correlated with acquired migration capabilities in response to OPN, also in the case of smaller primary tumors. Noteworthy, ICOSL is highly expressed in invasive breast carcinoma with poor prognosis[31] and several primary tumors, including breast cancer, acquire OPN expression at the step of tissue invasion and metastatic dissemination. Moreover, the present results suggest that ICOSL and OPN expression may build up an autocrine loop, since forced expression of ICOSL in 4T1 cells correlated with increased production of OPN in these cells.

It has recently been shown that the metastatic potential of 4T1 cells is greatly reduced in OPN−/− mice, whereas primary tumor growth is unaffected[32]. This shows that stromal-derived OPN is involved in metastatic dissemination of these cells, but provides no information concerning the effect on ICOSL, which is not expressed in wild-type 4T1 cells.

A recent report shows that ICOSL also binds to α_vβ_3 integrin, through an RGD motif that is located in the membrane-distal IgV domain of human ICOSL[33], whereas in the mouse, it is located in the membrane proximal to the IgC domain. This is intriguing, since α_vβ_3 integrin is a known ligand also of the RGD motif displayed by OPN, and it suggests that α_vβ_3 integrin may play a role in the complex interaction between ICOSL and OPN, and also in the conformational changes of OPN acting as a scaffold of multiple receptors on the cell surface.

In conclusion, this study shows that ICOSL is a receptor for OPN, interacting at a different domain of that used by ICOS. ICOSL binding by ICOS or OPN exerts opposing effects on cell migration, which is induced by OPN and dominantly inhibited by ICOS. The ICOS effect appears to be mediated by direct signaling through ICOSL, which has been demonstrated in several contexts, and which is capable of inhibiting cell migration induced by several chemotactic factors besides OPN. The OPN effect might be mediated by direct signaling through ICOSL, or by the integration of multiple signals triggered by OPN, acting as a scaffold on multiple cell receptors, including ICOSL, CD44, and several integrins. In the light of the emerging role of the ICOS/ICOSL/ OPN trio in cancer progression, inflammation, adaptive immunity, and bone metabolism, this trio might be targeted in various ways in order to influence these processes in several human diseases.

## Methods

**Cell lines and reagents**. The tumor-cell lines were grown in culture dishes as a monolayer in RPMI 1640 medium plus 10% fetal bovine serum (FBS) (Life

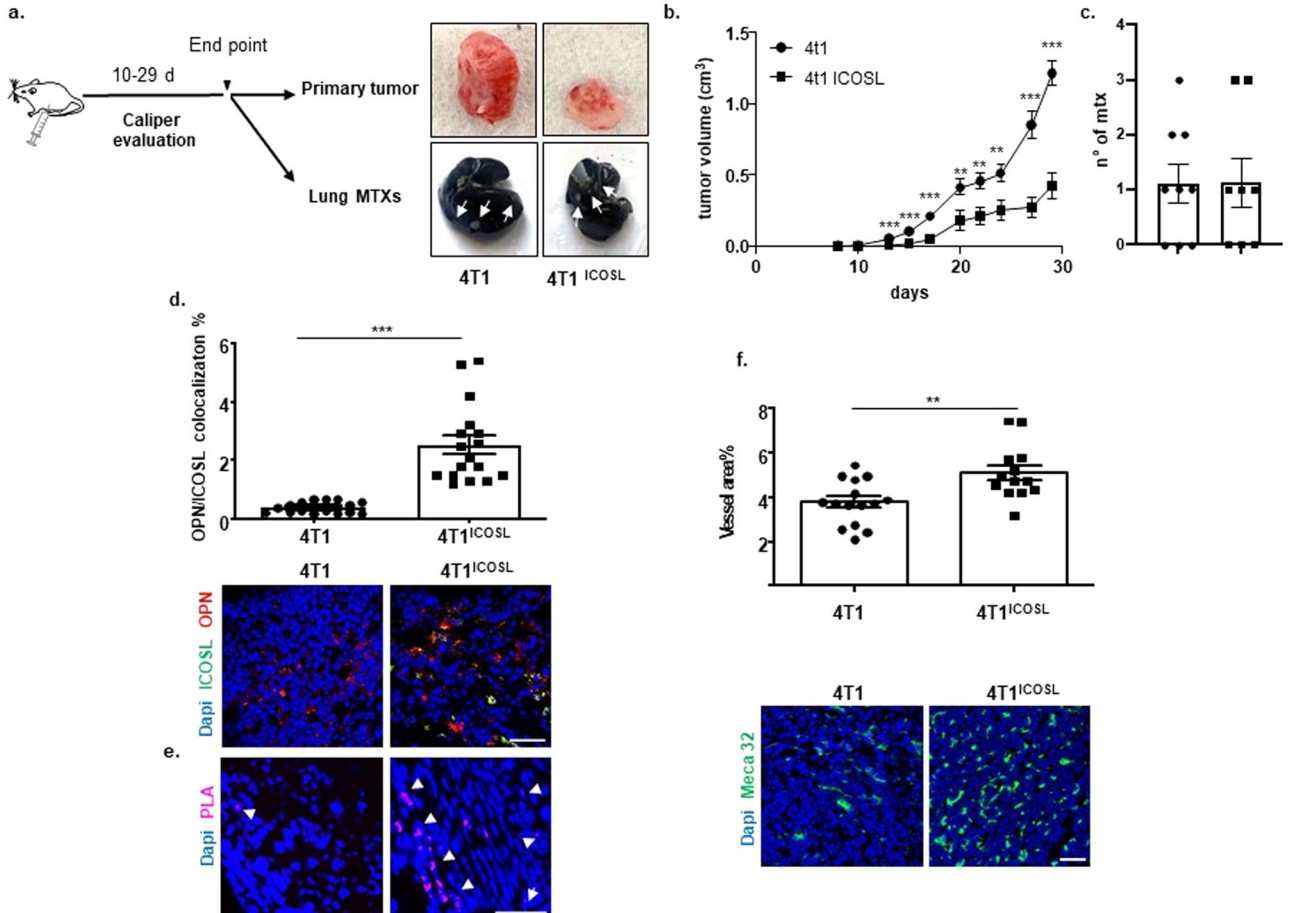

**Fig. 8 OPN/ICOSL binding modulates tumor growth, angiogenesis, and metastatization in vivo. a** Schematic illustration of experimental in vivo approach and images of excised 4T1and 4T$^{ICOSL}$ tumors, where white arrows indicate metastases. **b** Primary tumor growth of 4T1 cells in comparison with 4T$^{ICOSL}$ cells. $0.1 \times 10^6$ 4T1 and 4T1$^{ICOSL}$ cells were subcutaneously injected into the mammary fat pad of Balb/c mice ($n = 8$–9 biological replicate per group). Tumor growth was monitored using a caliper by measuring tumor size in 3 cross-directions. The Mann–Whitney test was used to compare differences in tumor growth between the two groups per day, $**P < 0.01$, $***P < 0.001$. **c** Numbers of 4T1 and 4T1$^{ICOSL}$ tumor lung metastases at the end point (day 29). **d** OPN and ICOSL protein level and localization were evaluated by confocal analysis. Images were quantified using ImageJ software which permits to calculate the ratio between red (OPN) and green (ICOSL) channels; values are expressed as percentage of red-green co-staining. Dot plot shows OPN and ICOSL fluorescence intensity colocalization (3–4 biological replicate) and the Mann–Whitney test was used. **e** PLA on primary tumor specimens showing OPN–ICOSL interaction. **f** As assessed by confocal analysis of Meca32 immunostaining, vessel density in 4T1$^{ICOSL}$ tumors was significantly increased (by 26%) compared with wild-type 4T1. The percentage of surface area occupied by vessels was quantified as Meca32 positive staining. Results have been analyzed using unpaired Mann–Whitney $U$ test. Scale bars: 50 μm ($n = 3$–4 biological replicate). Four images at a magnification of ×40 were analyzed for each sample, considering three mice per treatment group. All data are expressed as means ± standard error.

technologies, Carlsbad, California, USA) 100 U ml$^{-1}$ penicillin, and 100 μg ml$^{-1}$ streptomycin at 37 °C in 5% $CO_2$-humidified atmosphere. The following tumor-cell lines were used: JR8, M14, PCF2, A2058 (human melanoma) Rajj (human Burkitt's lymphoma) and 4T1 (mouse breast cancer) purchased from the American Type Culture Collection (ATCC; Virginia, USA).

HUVECs were isolated from human umbilical veins by trypsin treatment (1%) and cultured in M199 medium (Life technologies, Carlsbad, California, USA), with the addition of 20% FBS, 100 U ml$^{-1}$ penicillin, 100 μg ml$^{-1}$ streptomycin, 5 UI ml$^{-1}$ heparin, 12 μg ml$^{-1}$ bovine brain extract, and 200 mM glutamine. HUVECs were grown to confluence in flasks and used at passages two through five. The use of HUVEC was approved by the Ethics Committee of the "Presidio Ospedaliero Martini" of Turin and conducted in accordance with the Declaration of Helsinki. Written informed consent was obtained from all donors. Cells were routinely screened for the mycoplasma contamination by PCR. All mycoplasma tests performed during this study showed negative.

cDNAs coding for the human and murine OPN were cloned by PCR into a pUCOE expression vector as described elsewhere[12] and stably transfected into Chinese hamster ovary cells (CHOs). Cell supernatants were collected, and the recombinant proteins were purified on a nickel-nitrilotriacetic agarose resin (Qiagen, Limburg, Netherlands) and characterized by Western blotting using either an antibody directed against the His tag (Tetra-His Antibody, Qiagen, Valencia, CA, USA) or an anti-OPN polyclonal antibody (Millipore, Billerica, MA, USA).

cDNA human GST-OPN was cloned by PCR into p4t1 prokaryotic expression vector using the following primers: OPN-GST-for: 5′-CCGGATCCCATACCAGTTAAACAGGCTG-3′, OPN-GST-rev: 5′-CCCCTCGAGAATAGATACACATTCAACC-3′.

The OPN-GST fusion protein was transformed in *Escherichia coli* BL21 (DE3). Bacteria were precultured overnight in 2xTY medium and diluted in 1 liter of the same medium. The optical density (OD) was constantly monitored until it reached an OD of 1.5. Protein expression was induced by the addition of 0.5 mM isopropyl 1-thio-β-D-galactopyranoside and following 90 min incubation at 37 °C. The cells were then pelleted and resuspended in 30 ml of PBS, pH 7.4 and lysed following eight cycles of sonication. Pellet and supernatant were separated by centrifugation, and the supernatant was applied to a pre-equilibrated Glutathione Sepharose column (GE Healthcare) and incubated for 1 h at 4 °C. The resin was washed thoroughly with PBS, pH 7.4 10 mM GSH, and protein was eluted with buffer PBS, pH 7.4 60 mM GSH.

**Animals**. For this study, adult female mice from the following four standard inbred strains were used: C57BL/6 J (The Jackson Laboratory, Bar Harbor, ME, USA); B6.129S6(Cg)-Spp1$^{tm1Blh}$/J, also known as OPN−/− (The Jackson Laboratory); B6.129P2-Icos<tm1Mak>/J also know as ICOS−/− (The Jackson Laboratory), BALB/cOlaHsd, or BALB/c mice (Envigo, Huntingdon, UK); and MRL/MpJ-Fas$^{lpr}$/J,

commonly known as MRL-lpr (The Jackson Laboratory). Mice of the first four strains were used at 8–10 weeks of age, while those of the latter strain were 12 weeks old. All animals were maintained under pathogen-free conditions in the animal facility of Università del Piemonte Orientale; they were fed ad libitum on rodent chow and water was freely available in the home cages, the ambient temperature was maintained at 21 ± 1 °C. All experimental procedures were conducted during the light phase of a 12:12-h light:dark cycle. Experimental procedures were conducted, following European guidelines, in accordance with both the University Ethical Committee and the National Institutes of Health Ministry and Care Committee (protocol number DB064.42), both of which approved the protocol.

**Cell migration assay**. In the Boyden chamber (BD Biosciences, San Jose, CA, USA) $8 \times 10^3$ cells were plated onto the apical side of 50 mg ml$^{-1}$ Matrigel-coated filters (8.2-mm diameter and 0.5-mm pore size; Neuro Probe; BIOMAP snc, Milan, Italy) in serum-free medium with or without 2 µg ml$^{-1}$ of ICOS-Fc (Bio-Techne, Minneapolis, Minnesota, USA). Medium containing 20% FCS or 10 µg ml$^{-1}$ rOPN (Bio-Techne, Minneapolis, Minnesota, USA) was placed in the basolateral chamber as a chemoattractant for the tumor, while 10 ng ml$^{-1}$ VEGF-α as a chemoattractant for HUVECs. The chamber was incubated at 37 °C under 5% CO$_2$. After 8 h, cells on the apical side were wiped off with Q-tips. Cells on the bottom of the filter were stained with crystal violet and all counted (quadruplicate filters) with an inverted microscope (magnification ×100). Data are shown as percentages of treated cell migration versus control migration measured for untreated cells.

**ICOSL cloning**. RNA from Raji cell line (ATCC, Manassas, Virginia, USA) was extracted using TRI reagent (Merck, Darmstadt, Germany). 1 µg of RNA was retrotranscribed to cDNA using the QauntiTect Reverse Transcription Kit (Qiagen, Hilden, Germany). cDNA of ICOSL was amplified using PCR with specific oli-gonucleotides (ICOSL-For: 5′-GATGCTAGCATGCGGCTGGGCAGTCCTG-3′ and ICOSL-Rev: 5′-CTTAAGCTTTCAAACGTGGCCAGTGAGCTC-3′). After PCR, the amplicon was purified and cloned into pcDNA 3.1 vector (Life tech-nologies, Carlsbad, California, USA). To generate ICOSL without intracytoplasmic domain (ICOSL-TL), the following primers were used, annealing to the ICOSL-pcDNA expression plasmid (ICOSL-TL-for: 5′-ATAGGATCCCGCCACCATG CGGCTGGGCAGTCCTG-3′ and ICOSL-TL-rev: 5′-CCGGAATTCCGCCACCA TGCGGCTGGGCAGTCCTG-3′). To generate ICOSL without extracellular domain (ICOSL-intracytoplasmic (ICOSL-IC)), the following primers were used (ICOSL-IC-for: 5′-CATCACCATTGGAGCATCCTGGCTGTCC-3′ and ICOSL-IC-rev: 5′-GTGATGGTGATCAGCTCGAAGGCTGCTG-3′). cDNA derived from RNA extracted from mouse splenocytes was obtained using the above protocol. cDNA of mouse ICOSL (mICOSL) was generated using the following primers (mICOSL-for: 5′-GATGCTAGCATGCAGCTAAAGTGTCCCTG-3′ and mICOSL-rev: 5′-CTTAAGCTTTCAGGCGTGGTCTGTAAGTTG-3′) and cloned into pcDNA 3.1 vector. All constructs were transformed into JM109 bacteria (Promega, Madison, Wisconsin, USA) and the resulting colonies were sequenced by Sanger sequencing (Life technologies, Carlsbad, California, USA).

**Cell transfection**. A2058, 4T1 and HeLa cells were plated in 10 cm dish plates (10$^6$ cells/plate) for 24 h in complete medium RPMI-1640 (Lonza, Basel, Switzerland) (A2059) or DMEM (Lonza, Basel, Switzerland) (Hela and 4T1). Cells were then transfected with 10 µg of plasmid DNA and 30 µl of lipofectamine® 3000 (Life technologies, Carlsbad, California, USA) and expression of ICOSL was evaluated, after 48 h either with real-time PCR, using a specific probe for ICOSL, by flow cytometry using PE-conjugated antibodies to human ICOSL (Bio-Techne, Min-neapolis, Minnesota, USA), or by immunofluorescence using primary antibodies to mouse ICOSL (Life technologies, Carlsbad, California, USA) followed by Alexa Flour 555 secondary antibodies (Anti-goat, Life technologies, Carlsbad, California, USA).

**ICOSL silencing**. HUVECs were seeded (1.5 ×10$^5$ well$^{-1}$) on 6 well plates for 24 h, and then ICOSL was silenced using lipofectamine® RNAiMAX transfection reagent (Life technologies, Carlsbad, California, USA) with two siRNA targeting two dif-ferent exons of ICOSL (oligo1: ICOSLGHSS177318 (5′-CAGCAGCCUUCGAG-CUGAUACUCAG-3′ and 5′-CUGAGUAUCAGCUCGAAGGCUGCUG-3′) and oligo2: ICOSLGHSS118565 (5′-GGCCCAACGUGUACUGGAUCAAUAA-3′ and 5′-UUAUUGAUCCAGUACACGUUGGGCC-3′) (Life technologies, Carlsbad, California, USA).

Real-time PCR was used to evaluate the expression of ICOSL after silencing. After 24 h, HUVECs were starved and RNA was isolated using TRI reagent (Merck, Darmstadt, Germany). 1 µg of RNA was retrotranscribed to cDNA using the QauntiTect Reverse Transcription Kit (Qiagen, Hilden, Germany). ICOSL expression was evaluated with a gene expression assay amount. Real-time PCR was performed using CFX96 System (Bio-Rad Laboratories, Hercules, California, USA), samples were run in duplicate in a 10 µl final volume containing 1 µl of diluted cDNA, 5 µl of TaqMan Universal PCR Master Mix (Life technologies, Carlsbad, California, USA), and 0.5 µl of Assay-on Demand mix. The results were analyzed with a ΔΔ threshold cycle method.

**ELISA-based interaction assay**. OPN (60 nM) (Bio-Techne, Minneapolis, Min-nesota, USA) in PBS was used to coat Nunc MaxiSorp™ flat-bottom plates (Life technologies, Carlsbad, California, USA) overnight at 4 °C. After one wash with PBS + 0.25% Triton X-100 (Merck, Darmstadt, Germany), 3% of Bovine Serum Albumin (BSA) (Merck, Darmstadt, Germany) in PBS + 0.05% Tween-20 (Merck, Darmstadt, Germany) was added for 1 h at 25 °C. After three washes, the plate was incubated with titrated amounts (from 60 nM to 1.8 nM) of ICOSL-Fc (Bio-Techne system, Minneapolis, Minnesota, USA) in PBS + 0.05% Tween-20 for 1 h at 25 °C with or without 60 nM of ICOS-Fc (competition assay). After washing, ICOSL-Fc binding to OPN was evaluated using HRP conjugated anti-human-IgG1 antibodies (1:4000) (Dako, Santa Clara, California, USA) in PBS + 0.05% Tween-20 for 1 h at 25 °C, followed by washing and addition of the TMB substrate (Merck, Darmstadt, Germany). The reaction was stopped after 2 min with H$_2$SO$_4$ 2 N (Merck, Darmstadt, Germany) and absorbance was assessed at 450 nm using a Victor-X1 plate reader (Perkinelmer, Waltham, Massachusetts, USA).

For the reciprocal ELISA, 60 nM of ICOSL-Fc in PBS was used to coat the ELISA plates and titrated amounts of OPN (from 60 nM to 1.8 nM) were incubated in the wells to assess binding. After washing, the bound OPN was revealed using biotinylated polyclonal anti-OPN antibodies (Bio-Techne, Minneapolis, Minnesota, USA) with an incubation of 2 h at 25 °C; after three washes Streptavidin-HRP (Bio-Techne, Minneapolis, Minnesota, USA) was added for 20 min at 25 °C; the signal was developed following the above protocols.

In other experiments, 60 nM of OPN-GST, OPNb or home-made OPN-FL were used to compare the interaction between commercial OPN and the others, and 60 nM of OPN-N terminal and OPN-C terminal respectively were used to map the interaction site of ICOSL-OPN following the below protocol.

**Pull-down assay**. ICOSL-Fc and poly-his-OPN (10 µg each; Bio-Techne system, Minneapolis, Minnesota, USA) were co-incubated in PBS (1 ml) at room tem-perature (r.t.) on rotation for 1 h. Then ICOSL-Fc was precipitated using Sepharose-protein G (GE Healthcare, Chicago, Illinois, USA). Sample buffer with 20% of β-mercaptoetanhol (Merck, Darmstadt, Germany) was used to dissociate the proteins, which were then run in SDS/PAGE followed by Western blotting using 1 µg ml$^{-1}$ of anti-OPN polyclonal antibodies (Bio-Techne system, Minnea-polis, Minnesota, USA).

**Limited Proteolysis experiments**. Limited Proteolysis experiments were carried out on commercial recombinant His-tagged Osteopontin (OPN), produced by R&D (Minneapolis, Minnesota, US) in a mouse myeloma cell line, containing the physiological post-translational modifications but lacking 15 aa of exon 5, on ICOSL-Fc provided by R&D (Minneapolis, Minnesota, US) and on OPN/ICOSL-Fc. The three samples were treated with trypsin and chymotrypsin (SigmaAldrich) in phosphate Dulbecco's buffer (pH = 7.4), following set-up experiments to opti-mize enzyme:substrate (E:S) ratios for each protein. For both proteases, the E:S ratio was 1:5000 for both isolated OPN and OPN/ICOSL-Fc complex and 1:100 for isolated ICOSL-Fc. In the complex analysis, OPN and ICOSL-Fc were pre-incubated with 2:1 molar ratio for 1 h at r.t. before starting the proteolysis experiment, in order to allow complex formation. Enzymatic digestions were run for 30 min, after which 2% TFA was added to halt the reaction. The proteolysis mixtures were resolved on a 15% acrylamide SDS-PAGE gel, and differential bands were cut and subjected to in situ hydrolysis with trypsin, following the protocol reported in Medugno et al., 2003[22]. Tryptic peptide mixtures were analyzed both by 4800 MALDI TOF/TOF Analyzer (Sciex, Framingham, Massachusetts, USA) and by nanoLC-MS/MS using a Proxeon-nanoEasy II-LTQ Orbitrap XL (Life tech-nologies, Carlsbad, California, USA). In this latter, peptide fractionation was per-formed on a C18 capillary reverse-phase column (200 mm, 75 µm, 5 µm) working at 250 nl min$^{-1}$ flow rate, using a step gradient of eluent B (0.2% formic acid, 95% acetonitrile LC-MS Grade) from 10 to 60% over 69 min and 60 to 95% over 3 min. Mass spectrometric analyses were performed using data-dependent acquisition (DDA) mode over the 400 to 1800 $m z^{-1}$ range, followed by acquisition in MS/MS mode of the five most abundant ions present in each MS scan. Peptides were identified using MASCOT software (Matrix Science Boston, USA) searching in a database containing only OPN and ICOSL-Fc sequences.

**Cross-linking experiments**. In-house purified GST-OPN (OPNa acccession number NP_001035157.1) and His-tagged ICOSL (Bio-Techne, Minneapolis, Minnesota, US) were pre-incubated at 2:1 molar ratio for 1 h at r.t., as described above, and once the complex was formed the DTSSP cross-linker (3,3′-dithiobis (sulfosuccinimidyl propionate, Life technologies, Carlsbad, California, USA) was added in a molar excess of 1:50 and incubated for 30 min at 4 °C. The reaction mixtures were loaded onto 8% non-reducing SDS-PAGE gel. Bands corresponding to isolated proteins and to the complex were excised from the gel and subjected to in situ hydrolysis with trypsin, skipping the reduction and alkylation steps, to preserve DTSSP disulfide integrity. Peptide mixtures were analyzed by MALDI-MS and by nanoLC-MS/MS, as described above. Peptide identification was carried out both manually and using software such as MASCOT and Batch-Tag, MS-Bridge and MS-Product tools within the Protein Prospector package. An aliquot of each peptide mixture was reduced in 20 mM ditiothreitol for 2 h at 37 °C and then alkylated with 55 mM of iodoacetamide for 30 min in the dark at r.t.; the mixtures

thus obtained were analyzed by MALDI-MS and nanoLC-MS/MS, as described above.

**4T1 breast cancer mouse model**. To evaluate tumor growth and tumor metastatization, $1 \times 10^5$ 4T1 and 4T1$^{ICOSL}$ cells were injected into the fat pad of 8–10-week-old BALB/c female mice (Envigo, Huntingdon, UK). Tumor growth was evaluated using a caliper, starting when the tumors became palpable, every two days for 29 days from cell injection. At the end point, the mice were anesthetized to allow intracardiac collection of the blood, then sacrificed. Superficial pulmonary metastases were contrasted by black India-ink intratracheal infusion and counted on explanted lung lobes under a stereoscopic microscope. The numbers of lung metastases were counted by two observers working blind.

**Immunofluorescence on animal specimens**. After death, all animals underwent complete necropsy. Kidney, spleen, liver, lung, heart, and brain were weighed and macroscopically analyzed. Immediately after dissection, only kidney and tumors were embedded in OCT compound (Killik, Bioptica, Milan, Italy), and stored at −80 °C until use. 10 μm sections were cut using a cryostat (Leica). For immunofluoresce analysis sections were air-dried, fixed in zinc fixative (6.05 g Tris, 0.35 g Ca (C2H3O2)2, 2.5 g Zn (C2H3O2)2, 2.5 g ZnCl, 3.8 ml HCl 37%) for 10 min at r.t. and permeabilized with PBS + 0.1 % Triton X-100 (Merck, Darmstadt, Germany) for 5 min at r.t. Tissues were then blocked with 3% BSA and 3% donkey serum in PBS (blocking solution) for 1 h at r.t. After quickly washing in PBS 1x, tissues were incubated either for 1 h at r.t. or overnight at 4 °C with the following primary antibodies: rabbit polyclonal anti-ICOSL (Antibodies online, Aachen, Germany), goat polyclonal anti-OPN (Abcam, Cambridge, UK) or rat monoclonal anti-panendothelial cell antigen (clone Meca32, BD Pharmingen), all diluted 1:100 in blocking solution. After three washing in PBS 1x, tumors were incubated for 1 h at r.t. with the following secondary antibodies: Alexa 555 donkey anti-goat and Alexa 488 donkey anti-rabbit or Alexa 488 donkey anti-rat (Life technologies, Carlsbad, California, USA), diluted 1:400 in PBS.

Sections were then rinsed three times in PBS 1× and stained with 0.5 mg ml$^{-1}$ of the fluorescent dye 4,6-diamidino-2-phenylindole-dihydrochloride (DAPI) (Merck, Darmstadt, Germany) for 5 min, to color the cell nuclei. Slides were mounted using Prolong anti-fade mounting medium (Slow Fade AntiFADE Kit, Molecular Probes, Invitrogen). Images of the sections were acquired with a Leica TCS SP2 AOBS confocal laser-scanning microscope (Leica Microsystems) equipped with a ×20 or ×40 objective lens. Images were quantified using ImageJ software which permits to calculate the ratio between red (OPN) and green (ICOSL) channel; values are expressed as percentage of red-green co-staining. Four images at a magnification of ×40 were analyzed for each sample, considering three mice per treatment group.

**Tumor vasculature quantification**. Tumor vasculature was quantified by means of ImageJ software as the area occupied by Meca32-positive structures, compared with the total tissue area. Four images at a magnification of ×20 were analyzed for each sample, considering three mice per treatment group.

**Proximity ligation assays (PLA)**. PLA is a powerful tool that allows in situ detection of protein interactions with high specificity and sensitivity. This technique uses one pair of primary antibodies, raised in different species, to target proteins of interest whose interaction is to be studied. The kit includes a pair of oligonucleotide-labeled secondary antibodies (PLA probes), ligation and DNA polymerase. If the proteins of interest are closer than 40 nm, the secondary oligos couple and the fluorochromes hybridize; after signal amplification, the result is then viewable as discrete spots.

The concentrations for immunostaining of anti-OPN and ICOSL primary antibodies were previously optimized as described above. Kidney tissues from C57Bl/6 J mice and breast tumors from 4T1 were firstly processed as described above for immunofluorescence and incubated with anti-OPN and anti-ICOSL overnight at 4 °C. PLA analysis was then carried out using the Duolink® In Situ Far Red Starter Kit Goat/Rabbit (# DUO92008) according to the manufacturer's protocol (Merck, Darmstadt, Germany). Additional reagents were used: Duolink® In Situ PLA® Probe Anti-Goat MINUS (# DUO92006, Merck) and Duolink® In Situ PLA® Probe Anti-Rabbit PLUS (# DUO92002, Merck), Duolink® In Situ Wash Buffers, Fluorescence (# DUO82049, Merck) and Duolink® In Situ Mounting Medium with DAPI (# DUO82040, Merck). Images were acquired with a Leica TCS SP2 AOBS confocal laser-scanning microscope (Leica Microsystems).

**Statistics and reproducibility**. Details on statistical analyses and number of biological replicates (n) can be found in the figure legends. Experiments were set up with, at least, three replicates for each sample unless otherwise stated. *P values* below 0.05 were considered statistically significant. The statistical analyses were performed with GraphPad Instat software (GraphPad Software). All data sources underlying the figure graphs, are included in Supplementary Data 1.

**Reporting summary**. Further information on research design is available in the Nature Research Reporting Summary linked to this article.

## Data availability
The authors declare that the data supporting the findings of this study are included within the article or supplementary files and are available from the corresponding author upon reasonable request. New plasmids reported in this study (huICOSL-FL, huICOSL-TL, huICOSL-IC, Ms ICOSL, huGST-OPN) will be available via Addgene.

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

## Acknowledgements

The authors are grateful to the Obstetrics and Gynaecologic Unit, Martini Hospital, Torino, for the providing human umbilical cords and to Davide Ferraris for technical support. This work was supported by the Italian Ministry of Education, University and Research (MIUR) program "Departments of Excellence 2018–2022", FOHN and AGING Projects, Fondazione Cariplo 2019–3277 to A.C., the Associazione Italiana Ricerca sul Cancro (IG 20714, AIRC, Milano), Fondazione Amici di Jean (Torino), and Fondazione Cariplo (2017–0535) to U.D., National Ministry of University and research PRIN 2017 (grant 201799WCRH) to G.B., Consorzio Interuniversitario di Biotecnologie (CIB) bando "Network-CIB: Catalisi dell'Innovazione nelle biotecnologie" to G.B.

## Author contributions

D.R., C.D., G.C., G.B. performed in vitro experiments, analyzed data; F.M. and N.C. performed in vivo experiments and tissue analysis; II carried out conformational experiments and mass spectrometry analyses; E.B. and C.L.G. provided ICOS-Fc molecules; R.B. analyzed tissues staining, M.M. and L.B. designed and analyzed limited proteolysis and cross-linking data; GBal constructed the model; J.M.R. critically read the manuscript; U.D., C.D., and A.C. conceived the project and wrote the manuscript.

## Competing interests

A.C., U.D., C.D., E.B., and C.L.G. are listed as inventors on the patent PCT/IB2019/050154 "Novel anti-tumor therapeutic agents". E.B., U.D., and C.L.G. are founders of an UPO Spinoff (NOVAICOS). All other authors declare no competing interests.
