## [Peer Review File · Communications Biology]

Reviewers' comments:

Reviewer #1 (Remarks to the Author):

see attached

Reviewer #2 (Remarks to the Author):

This paper provides a good investigation on the interaction of ICOSL and OPN which has not been identified/reported by other researchers. The novelty is good and the data volume is significant. However, there are few important points need to be clarified/answered by the authors before this manuscript could further consider for publication:

1. In Figure 1e, could the authors kindly provide more explanation/justification on why the HUVEC cells migration experiments performed in the paper were using VEGF- α instead of 20% FBS. In addition, in Figure 1e, if the authors added VEGF- α in ICOSL silenced group without adding OPN, will the migration effect able to restore? If yes, how to explain this phenomenon? And what is the control cell type used in the white bar in Figure 1e, is it Scr? Please kindly clarify.
2. In Figure 6B, I think the representative colony figures shown are not correct, at least the results are not consistent with the bar shown. Please kindly clarify.
3. In Figure S2a, the labelling of the right bar chart is wrong, should be relatively expression of OPN/GAPDH. Please amend accordingly.
4. It seems that the in vitro work indicated that the ICOSL increase the metastasis depend on OPN, but in vivo work didn't show any significant effect on this hypothesis as the authors described. However, the elucidation of the conclusion regarding the in vivo work part (Figure 7) are quite blur. Please provide a clear statement/claim. In addition, any difference on weight loss was observed during the in vivo work period? Please kindly clarify.

In “Osteopontin binds ICOSL promoting tumor metastatization” by Raineri & colleagues, the authors describe a novel interaction for the secreted form of osteopontin (OPN) with ICOSL, a member of the B7 family of co-stimulatory receptors implicated in cancer development and metastasis. They find that OPN directly interacts with ICOSL at a novel binding site described in this manuscript, and acts to promote cell migration through signaling requiring both the extracellular binding and intracellular signaling domains of ICOSL. The authors then find that this OPN-ICOSL signaling disrupts attachment-independent division in an ICOSL expressing tumor cell line, which correlates with a reduction in tumor growth. Overall, the manuscript provides a novel and valuable insight into the complex biology of osteopontin in a technically sound manner, although there are several issues that should be addressed.

Strengths

1. The major finding in this paper, that OPN directly binds ICOSL and induces a cellular migration program, is supported by multiple lines of solid evidence converging on this hypothesis.
2. The authors provide some evidence for the structural conformation of the OPN-ICOSL complex, which offers insight into how the soluble form of OPN might act as a scaffold for surface receptors. Given that OPN is an intrinsically disordered protein, determining structural details of its interaction with a candidate ligand is not straightforward, and the authors should be commended for this discovery.
3. The authors make a point to test several variant sources of OPN, including commercial (OPN-b), bacterially-sourced (no PTMs), and a full-length protein produced in-house. All three of these preparations show similar ICOSL binding, which is a solid confirmation that the finding presented here is not due to technical artifacts introduced by different manufacturing

procedures and greatly increases the chance that other groups will be able to use these findings in their own work.

Major Issues

1. The thesis that OPN-ICOSL interaction promotes tumor metastases is not fully supported by these data. Although there is a) a statistically detectable increase in vessel density (Figure 7f), that is correlated with angiogenesis (and metastatic disease), and b) that the OPN-ICOSL expressing tumor cells make a tumor front in Figure 7e, these findings are not sufficient to infer that the tumor cells themselves are more metastatic.

The authors might wish to provide additional data supporting the thesis that OPN-ICOSL interactions can promote metastatic disease or include a more cautious interpretation of these data in the discussion.

Minor Issues

1. Figure 1a-b: It would be helpful to provide flow cytometry staining of ICOSL levels on M14, JR8, PCF2, A2058, and the A2058 transfected overexpressing cells to support this figure. This should not be difficult, since similar staining is provided for transfected HeLa in Figure 1g.
2. Figure 2a: the authors mention in the text that a reciprocal ELISA was performed, where ICOSL-Fc is the capture protein against increasing concentrations of OPN-b. This is highlighted in the discussion as stronger evidence that OPN and ICOSL interact. However, the reciprocal ELISA is not shown. This should be included, at least as a supplemental figure.
3. Figure 2c: The authors might include a more explicit rationale for the use of the MLR/lpr mouse model to study OPN-ICOSL interactions. For readers unfamiliar with this model, it would be useful to provide some background for its use.

4. Figure 3a: although the lanes in this figure are numbered and explained in the legend, it would be helpful to instead label them directly on the figure, especially since given the present layout, lanes 4 and 5 look like they have been Chymotrypsin treated when in fact they are untreated.

Something like this is suggested:

	Trypsin				Chymotrypsin			
M	OPN	OPN+ICOSL	ICOSL	OPN	ICOSL	OPN	OPN+ICOSL	ICOSL
	1	2	3	4	5	6	7	8

Copy-editing Issues:

1. Results, bottom of first paragraph: ICOSL did not respond to OPN (Fig. 1E)
2. Results, second paragraph: HeLa transfected for full-length ICOSL (ICOSL)
3. Figure 1e: y-axis label is rendered strangely and hard to read
4. Figure S2a: wrong y-axis label for OPN panel

Reviewer #1

In “Osteopontin binds ICOSL promoting tumor metastatization” by Raineri & colleagues, the authors describe a novel interaction for the secreted form of osteopontin (OPN) with ICOSL, a member of the B7 family of co-stimulatory receptors implicated in cancer development and metastasis. They find that OPN directly interacts with ICOSL at a novel binding site described in this manuscript, and acts to promote cell migration through signaling requiring both the extracellular binding and intracellular signaling domains of ICOSL. The authors then find that this OPN-ICOSL signaling disrupts attachment-independent division in an ICOSL expressing tumor cell line, which correlates with a reduction in tumor growth. Overall, the manuscript provides a novel and valuable insight into the complex biology of osteopontin in a technically sound manner, although there are several issues that should be addressed.

Strengths

- The major finding in this paper, that OPN directly binds ICOSL and induces a cellular migration program, is supported by multiple lines of solid evidences converging on this hypothesis.
- The authors provide some evidence for the structural conformation of the OPN-ICOSL complex, which offers insight into how the soluble form of OPN might act as a scaffold for surface receptors. Given that OPN is an intrinsically disordered protein, determining structural details of its interaction with a candidate ligand is not straightforward, and the authors should be commended for this discovery.
- The authors make a point to test several variant sources of OPN, including commercial (OPN-b), bacterially sourced (no PTMs), and a full-length protein produced in-house. All three of these preparations show similar ICOSL binding, which is a solid confirmation that the finding presented here is not due to technical artefacts introduced by different manufacturing procedures and greatly increases the chance that other groups will be able to use these findings in their own work.

We thank Reviewer #1 for his/her comments on the novelty of our data. Compatibly with COVID outbreak, we have worked to address her/his comments.

Major Issues

1. The thesis that OPN-ICOSL interaction promotes tumor metastases is not fully supported by these data. Although there is a) a statistically detectible increase in vessel density (Figure 7f), that is

correlated with angiogenesis (and metastatic disease), and b) that the OPN-ICOSL expressing tumor cells make a tumor front in Figure 7e, these findings are not sufficient to infer that the tumor cells themselves are more metastatic. The authors might wish to provide additional data supporting the thesis that OPN-ICOSL interactions can promote metastatic disease or include a more cautious interpretation of these data in the discussion.

Even though we considered the data on tumor angiogenesis very solid and clear, we agree with the reviewer that the results on the role of OPN/ICOSL interaction on metastatization are not conclusive. In agreement with Reviewer #1 we modified the discussion smoothing the speculation of OPN-ICOSL interaction on the metastatic process (lines 413-420).

Minor Issues

1. Figure 1a-b: It would be helpful to provide flow cytometry staining of ICOSL levels on M14, JR8, PCF2, A2058, and the A2058 transfected overexpressing cells to support this figure. This should not be difficult, since similar staining is provided for transfected HeLa in Figure 1g.

As suggested by the reviewer, we are including the flow cytometry staining images in the supplementary figures section (figure S.1) and we are referring to them within the text (lines 109-110 and 117),

2. Figure 2a: the authors mention in the text that a reciprocal ELISA was performed, where ICOSL- Fc is the capture protein against increasing concentrations of OPN-b. This is highlighted in the discussion as stronger evidence that OPN and ICOSL interact. However, the reciprocal ELISA is not shown. This should be included, at least as a supplemental figure.

We thank the reviewer for its suggestion and in the revised manuscript we added the results of reciprocal ELISA as supplemental figure (figure S.3) and described them in lines 155-156.

3. Figure 2c: The authors might include a more explicit rationale for the use of the MLR/lpr mouse model to study OPN-ICOSL interactions. For readers unfamiliar with this model, it would be useful to provide some background for its use.

We chose MLR/lpr mice because they are a strain with a mutation in the FAS gene, leading over time to a spontaneous autoimmune and lymphoproliferative syndrome resembling autoimmune lymphoproliferative syndrome (ALPS) in the early life and mimicking human systemic lupus erythematosus in the later stages. Moreover, clinical features include an immune complexes-mediated glomerulonephritis with increased OPN levels in the kidney. Since, ICOSL is constitutively expressed at high levels in this organ, we aim to detect in the kidney, if present, any OPN and ICOSL interaction. Thus, MRL/lpr mice were employed as a positive control for PLA. This concept has been clarified in results, lines 174-177.

4. Figure 3a: although the lanes in this figure are numbered and explained in the legend, it would be helpful to instead label them directly on the figure, especially since given the present layout, lanes 4 and 5 look like they have been Chymotrypsin treated when in fact they are untreated. Something like this is suggested:

Trypsin			Chymotrypsin					
M	OPN	OPN+ICOSL	ICOSL	OPN	ICOSL	OPN	OPN+ICOSL	ICOSL
	1	2	3	4	5	6	7	8

We modified the layout of Figure 3a according to reviewer' suggestion.

Copy-editing Issues:

1. Results, bottom of first paragraph: ICOSL did not respond to OPN (Fig. 1E)

We thank the reviewer for picking up this typo and we corrected it accordingly.

2. Results, second paragraph: HeLa transfected for full-length ICOSL (ICOSL)

We thank the reviewer for picking up this inconsistency, and we corrected it accordingly.

3. Figure 1e: y-axis label is rendered strangely and hard to read.

We modified the Y-axis to make it clearer, according to reviewer' suggestion.

4. Figure S2a: wrong y-axis label for OPN panel

We apologize for this mistake and we modified accordingly.

REVIEWER #2

This paper provides a good investigation on the interaction of ICOSL and OPN which has not been identified/reported by other researchers. The novelty is good, and the data volume is significant. However, there are few important points need to be clarified/answered by the authors before this manuscript could further consider for publication:

We thank Reviewer #2 for her/his constructive suggestions. We have worked to address her/his comments.

1. In Figure 1e, could the authors kindly provide more explanation/justification on why the HUVEC cells migration experiments performed in the paper were using VEGF- α instead of 20% FBS. In addition, in Figure 1e, if the authors added VEGF-a in ICOSL silenced group without adding OPN, will the migration effect able to restore? If yes, how to explain this phenomenon? And what is the control cell type used in the white bar in Figure 1e, is it Scr? Please kindly clarify.

We commonly use VEGF as internal positive control, to test the capability of specific HUVEC preparations, to migrate toward a strong stimulus as previously published by us in Dianzani et al.¹. We use, instead, 20% FBS, with other cell lines. As suggested by the reviewer, we added in the revised manuscript, the effect of ICOSL silencing on VEGF-induced migration (new Figs 1 d and e). As described in lines 123-125, ICOSL silencing has no effect on VEGF-mediated migration.

2. In Figure 6b, I think the representative colony figures shown are not correct, at least the results are not consistent with the bar shown. Please kindly clarify.

We agree with the reviewer that the images can be misleading: this is mainly ascribable to the fact that histogram bars show the number of colonies obtained in 3 independent experiments, while the images are from one experiment only. Moreover, another important difference between histograms and images is that all the visible colonies have been counted and reported in histograms, independently from their dimension. On the contrary, the images show much better larger colonies while smaller ones, are scarcely visible in such a small enlargement. We clarified this point in the Figure legend lines 903-905.

3. In Figure S2a, the labelling of the right bar chart is wrong, should be relatively expression of OPN/GAPDH. Please amend accordingly.

We apologize for this mistake and we modified accordingly.

4. It seems that the in vitro work indicated that the ICOSL increase the metastasis depend on OPN, but in vivo work didn't show any significant effect on this hypothesis as the authors described. However, the elucidation of the conclusion regarding the in vivo work part (Figure 7) are quite blur. Please provide a clear statement/claim. In addition, any difference on weight loss was observed during the in vivo work period? Please kindly clarify.

As suggested by Reviewer 2 we modified the discussion including a more cautious interpretation of our data Lines 413-420. Concerning body weight loss, we did not observe any significant difference. A new sentence has been added in line 318-319 and figure S5 (lane 948-949) has been added as supplementary.

References

1. Dianzani C, *et al.* B7h triggering inhibits the migration of tumor cell lines. *J Immunol* **192**, 4921-4931 (2014).

REVIEWERS' COMMENTS:

Reviewer #1 (Remarks to the Author):

The authors have addressed this reviewer's comments/concerns satisfactorily and the revised ms is much improved.

Reviewer #2 (Remarks to the Author):

Overall, the novelty of the study is good. The data volume that the authors presented in the manuscript are significant and can support their conclusions. I have happy with the current revised version of the manuscript, but would like to suggest the authors to add some more background of OPN (for example):

Osteopontin -- a promising biomarker for cancer therapy. Journal of Cancer 8(12): 3173-2183

Response to referees.

Reviewer #1 (Remarks to the Author):

The authors have addressed this reviewer's comments/concerns satisfactorily and the revised ms is much improved.

Response to reviewer 1.

We thank the reviewer for his/her comment.

Reviewer #2 (Remarks to the Author):

Overall, the novelty of the study is good. The data volume that the authors presented in the manuscript are significant and can support their conclusions. I have happy with the current revised version of the manuscript, but would like to suggest the authors to add some more background of OPN (for example):

Osteopontin -- a promising biomarker for cancer therapy. Journal of Cancer 8(12): 3173-2183

Response to reviewer 2.

We thank the reviewer for his/her comment. As recommended, we added few lines about the role of OPN as a therapeutic target in the introduction section and cited the suggested reference. Due to word count constraints, we could not add more. We added this part: "Importantly, current research, indicates that OPN inhibition would be a good therapeutic approach to metastatic disease. In preclinical models, OPN knockdown using RNAi, aptamers, or antibodies, have shown to have active roles in cancer treatment. Nevertheless, only a small number of these findings translate into clinical practice¹" (lines 79-82).